# Subgraph Federated Learning via Spectral Methods

**Javad Aliakbari**[1]    **Johan Östman**[2]    **Ashkan Panahi**[1]    **Alexandre Graell i Amat**[1]

[1]Chalmers University of Technology    [2]AI Sweden

## Abstract

We consider the problem of federated learning (FL) with graph-structured data distributed across multiple clients. In particular, we address the prevalent scenario of interconnected subgraphs, where interconnections between clients significantly influence the learning process. Existing approaches suffer from critical limitations, either requiring the exchange of sensitive node embeddings, thereby posing privacy risks, or relying on computationally-intensive steps, which hinders scalability. To tackle these challenges, we propose FEDLAP, a novel framework that leverages global structure information via Laplacian smoothing in the spectral domain to effectively capture inter-node dependencies while ensuring privacy and scalability. We provide a formal analysis of the privacy of FEDLAP, demonstrating that it preserves privacy. Notably, FEDLAP is the first subgraph FL scheme with strong privacy guarantees. Extensive experiments on benchmark datasets demonstrate that FEDLAP achieves competitive or superior utility compared to existing techniques.

## 1 Introduction

Graph-structured data naturally arise in a wide variety of real-world scenarios, with nodes representing distinct entities and edges reflecting relationships among them. Illustrative examples include anti-money laundering, social networks, and supply chains.

For graph-structured data, graph neural networks (GNNs) [1–3] have demonstrated remarkable effectiveness in tasks such as drug discovery, social network analysis, and traffic prediction, by capturing both node and structural information. However, in many real-world scenarios, as in the examples above, graph data is distributed across multiple parties, hindering direct data sharing due to regulatory, privacy, or proprietary considerations. This has led to the emergence of federated learning (FL) [4] as a promising paradigm to harness globally distributed graph data while preserving local data privacy. A particularly common setting for graph-structured data is **Subgraph Federated Learning (SFL)** [5], where each client holds a disjoint subgraph of a globally connected graph.

Several SFL methods have been proposed [5–11], but most except [11] involve sharing node features or learned embeddings, raising **critical privacy concerns**. Furthermore, attaining robust predictive accuracy under limited information exchange remains challenging. This reflects the well-known accuracy–privacy–communication trilemma [12], where improving one aspect often comes at the expense of the others. More recently, [13] proposed FEDSTRUCT, an SFL method that avoids sharing sensitive features by leveraging global graph structure. Although FEDSTRUCT offers stronger privacy than earlier methods (as clients share significantly less information), it still involves sharing partial adjacency matrix information and node structure features, which can potentially leak information. In addition, it **lacks a formal privacy analysis** and demands considerable communication overhead.

**Our Contribution.** We tackle the challenge of SFL for node classification, where a large graph is partitioned into disjoint subgraphs held by different clients. We adopt the common setting considered in [13, 10], where clients know how their subgraphs connect to others, but neither the central server nor any client can access the internal features or edges of other subgraphs. This scenario naturally

39th Conference on Neural Information Processing Systems (NeurIPS 2025).

arises in real-world settings—for example in banking, where a bank records a transaction to a customer at another bank and thus knows the recipient's identifier (e.g., IBAN). In anti-money laundering applications, the assumption of known interconnections is standard [14]. Our contributions push the Pareto frontier of the accuracy–privacy–communication trilemma by enhancing privacy and reducing communication, without compromising predictive performance. Specifically:

- We propose FEDLAP, a SFL framework that leverages global graph structure information via Laplacian smoothing in the spectral domain to effectively capture inter-node dependencies across subgraphs. The framework comprises two phases: an offline phase, executed once, in which global graph structure information is exchanged and does not involve any model training, and an online (training) phase that reduces to standard FL, offering higher flexibility than existing methods. FEDLAP achieves **utility close to a centralized approach while preserving privacy**.

- We propose a decentralized version of the Arnoldi iteration for spectral decomposition that **substantially reduces the computational cost of FEDLAP, improving efficiency over prior frameworks and enabling scalability to large, sparse graphs**. Crucially, information is exchanged only once before training, and thereafter only model parameters are shared with the server, as in standard FL.

- We provide a **rigorous privacy analysis** of FEDLAP, demonstrating strong privacy of local subgraph data. FEDLAP is the first SFL framework with **formally-supported privacy guarantees**—unlike existing methods, which lack such guarantees.

- Through extensive experiments for semi-supervised classification, we show that FEDLAP achieves performance on par with or surpassing existing SFL methods, with reduced communication overhead, better scalability, and enhanced privacy. The code is available at this link.

## 2 Related Work

**Subgraph federated learning.** Relevant works include FEDSAGE+ [5], FEDNI [6] , FEDDEP [15], FEDPUB [11], FEDGCN [10], FEDCOG [9], and FEDSTRUCT [13]. FEDSAGE+, FEDNI, and FEDDEP address missing inter-client information by employing inpainting techniques to infer features or embeddings. However, these methods face a critical trade-off: accurate inpainting exposes sensitive information and undermines privacy, while poor inpainting fails to improve node classification. FEDPUB avoids inpainting through personalized aggregation strategies, mitigating privacy risks but sacrificing performance due to limited access to global structural information. FEDGCN and FEDCOG incorporate GNNs via secure aggregation methods to exploit structural information. Yet, FEDGCN reveals aggregated node features to neighboring clients and FEDCOG intermediate embeddings, violating privacy (see [13] and [16]). FEDSTRUCT stands out as the most privacy-preserving method, while achieving similar or superior performance to FEDGCN and FEDCOG. However, it lacks a formal privacy analysis, and is communication-intensive, limiting its scalability to very large graphs.

**Structural information in GNNs.** Incorporating structural information into GNNs significantly enhances their representation power [17, 18]. [17] introduces structure-aware aggregation functions that improve expressivity beyond traditional GNNs, while FEDSTAR [18] shares explicit structural information in a FL setup to boost local model accuracy. FEDSTRUCT [13] is the first work to leverage explicit structural information in SFL to enhance performance while preserving privacy.

**Laplacian smoothing.** Foundational works [19, 20] highlighted both the theoretical and practical advantages of integrating graph Laplacians into semi-supervised frameworks, emphasizing their role in preserving the underlying data relationships. Modern GNNs [21, 22] draw inspiration from Laplacian smoothing by employing message-passing mechanisms that aggregate information from neighboring nodes, effectively promoting local smoothness in the learned embeddings.

## 3 Preliminaries and Setup

**General notation.** For a matrix $M \in \mathbb{R}^{n \times r}$, we denote by $M_{ij}$ its $(i, j)$-th element. We represent a submatrix of $M$ that is restricted in rows by the set $\mathcal{I}$ by $M_{\mathcal{I},:}$ and a submatrix that is restricted in columns by the set $\mathcal{J}$ by $M_{:,\mathcal{J}}$. Hence, $M_{i,:}$ and $M_{:,i}$ denote the $i$-th row and $i$-th column of $M$, respectively. A submatrix of $M$ that is restricted in rows by the set $\mathcal{I}$ and in columns by the set $\mathcal{J}$ by $M_{\mathcal{I},\mathcal{J}}$. We define $[k] = \{1, \ldots, k\}$.

**Graph notation.** We consider an undirected graph $\mathcal{G} = (\mathcal{V}, \mathcal{E}, X, Y)$, where $\mathcal{V} = \{1, 2, \ldots, n\}$ is the set of $n$ nodes, $\mathcal{E} = \{(u, v) | u, v \in \mathcal{V}\}$ the set of $m$ edges, $X \in \mathbb{R}^{n \times d}$ the node feature matrix,

and $\boldsymbol{Y} \in \mathbb{R}^{n \times d_c}$ the label matrix. Let $\boldsymbol{x}_v \in \mathbb{R}^d$ be the feature vector of node $v$, $\boldsymbol{y}_v \in \{0,1\}^{d_c}$ its one-hot encoded label vector, and $\tilde{\mathcal{V}} \subseteq \mathcal{V}$ the subset of nodes that possess labels. The adjacency matrix of graph $\mathcal{G}$ is denoted by $\boldsymbol{A} \in \mathbb{R}^{n \times n}$, where $A_{uv} = 1$ if $(u,v) \in \mathcal{E}$ and 0 otherwise. We define the diagonal matrix of node degrees as $\boldsymbol{D} \in \mathbb{R}^{n \times n}$, where $D_{uu} = \sum_v A_{uv}$. Also, we denote by $\tilde{\boldsymbol{A}} = \boldsymbol{A} + \boldsymbol{I}$ the self-loop adjacency matrix, by $\hat{\boldsymbol{A}} = \tilde{\boldsymbol{D}}^{-1} \tilde{\boldsymbol{A}}$ the normalized self-loop adjacency matrix, where $\tilde{D}_{uu} = \sum_{v \in \mathcal{V}} \tilde{A}_{uv}$, and by $\bar{\boldsymbol{A}} = \sum_{l=1}^{L} \beta_l \hat{\boldsymbol{A}}^l$ the $L$-hop *combined* neighborhood adjacency matrix. The elements of $\bar{\boldsymbol{A}}$ reflect the proximity of two nodes in the graph, with $\beta_l$, $\sum_{l=1}^{L} \beta_l = 1$, determining the contribution of each hop. The graph Laplacian of $\mathcal{G}$ is $\boldsymbol{L}_{\mathcal{G}} = \boldsymbol{D} - \boldsymbol{A}$.

**Laplacian smoothing.** Laplacian smoothing is a graph-based regularization method that encourages similar representations for neighboring nodes via a Laplacian loss term. Specifically, the total loss can be expressed as $\mathcal{L} = \mathcal{L}_c + \lambda_{\text{reg}} \mathcal{L}_{\text{reg}}$, where $\mathcal{L}_c$ is the supervised loss defined over the labeled part of the graph, $\lambda_{\text{reg}}$ is a weighting factor, and $\mathcal{L}_{\text{reg}}$ is the Laplacian regularization term defined as

$$\mathcal{L}_{\text{reg}} = \sum_{u,v} A_{uv} \| f_{\boldsymbol{\theta}}(\boldsymbol{x}_u) - f_{\boldsymbol{\theta}}(\boldsymbol{x}_v) \|^2 = \text{Tr}\left( f_{\boldsymbol{\theta}}(\boldsymbol{X})^{\mathsf{T}} \boldsymbol{L}_{\mathcal{G}} f_{\boldsymbol{\theta}}(\boldsymbol{X}) \right)$$

Here, $f_{\boldsymbol{\theta}}(\cdot)$ denotes a neural network-based differentiable function. The regularization term $\mathcal{L}_{\text{reg}}$ ensures that connected nodes in the graph have similar feature representations, thereby leveraging the graph structure to propagate label information from labeled nodes to unlabeled nodes.

**Setup.** We consider a scenario where data is structured according to a *global graph* $\mathcal{G} = (\mathcal{V}, \mathcal{E}, \boldsymbol{X}, \boldsymbol{Y})$, which is distributed among $K$ clients such that each client owns a smaller *local* subgraph. We denote by $\mathcal{G}_i = (\mathcal{V}_i, \mathcal{V}_i^*, \mathcal{E}_i, \mathcal{E}_i^*, \boldsymbol{X}_i, \boldsymbol{Y}_i)$ the subgraph of client $i$, where $\mathcal{V}_i \subseteq \mathcal{V}$ is the set of $n_i$ nodes that reside in client $i$, referred to as *internal nodes*, for which client $i$ knows their features. $\mathcal{V}_i^*$ is the set of nodes that do not reside in client $i$ but have at least one connection to nodes in $\mathcal{V}_i$. We call these nodes *external nodes*. Importantly, client $i$ does not have access to the features of nodes in $\mathcal{V}_i^*$. Furthermore, $\mathcal{E}_i$ represents the set of edges between nodes owned by client $i$ (intra-connections), $\mathcal{E}_i^*$ the set of edges between nodes of client $i$ and nodes of other clients (interconnections), $\boldsymbol{X}_i \in \mathbb{R}^{n_i \times d}$ the node feature matrix, and $\boldsymbol{Y}_i \in \mathbb{R}^{n_i \times d_c}$ the label matrix for the nodes within subgraph $\mathcal{G}_i$, and we denote by $\tilde{\mathcal{V}}_i$ the set of nodes that possess labels.

**Federated learning.** The FL problem can be formalized as learning the model parameters that minimize the aggregated loss across clients,

$$\boldsymbol{\theta}^* = \underset{\boldsymbol{\theta}}{\arg\min} \ \ \mathcal{L}_c(\boldsymbol{\theta}) \triangleq \frac{1}{|\tilde{\mathcal{V}}|} \sum_{i=1}^{K} \mathcal{L}_i(\boldsymbol{\theta}) \qquad \text{with} \qquad \mathcal{L}_i(\boldsymbol{\theta}) = \sum_{v \in \tilde{\mathcal{V}}_i} \text{CE}(\boldsymbol{y}_v, \hat{\boldsymbol{y}}_v), \tag{1}$$

where CE is the cross-entropy loss function between the true label $\boldsymbol{y}_v$ and the predicted label $\hat{\boldsymbol{y}}_v$.

The model $\boldsymbol{\theta}$ is trained iteratively over multiple epochs. At each epoch, the clients compute the local gradients $\nabla_{\boldsymbol{\theta}} \mathcal{L}_i(\boldsymbol{\theta})$ and send them to the central server. The server updates the model through gradient descent, $\boldsymbol{\theta} \leftarrow \boldsymbol{\theta} - \lambda \nabla_{\boldsymbol{\theta}} \mathcal{L}(\boldsymbol{\theta})$, $\nabla_{\boldsymbol{\theta}} \mathcal{L}(\boldsymbol{\theta}) = \frac{1}{|\tilde{\mathcal{V}}|} \sum_{i=1}^{K} \nabla_{\boldsymbol{\theta}} \mathcal{L}_i(\boldsymbol{\theta})$, and $\lambda$ is the learning rate.

## 4  FEDLAP

In this section, we introduce the **FEDLAP framework** (illustrated in Fig. 1), designed to exploit graph structure for enhancing SFL while rigorously addressing privacy and communication challenges.

FEDLAP builds upon the key insights from FEDSTRUCT [13] (discussed in Appendix A), explicitly addressing its main limitations: (i) the need to compute a costly global matrix $\bar{\boldsymbol{A}} \in \mathbb{R}^{n \times n}$, significantly increasing communication cost and privacy risks; (ii) optimization of a large structure feature matrix $\boldsymbol{S} \in \mathbb{R}^{n \times d_s}$ during training, which demands extensive communication and exposes the gradients of $\boldsymbol{S}$ to all clients, thereby increasing privacy leakage; and (iii) absence of formal privacy guarantees.

We resolve these challenges using two complementary strategies:

- **FEDLAP (Section 4.1)** employs Laplacian smoothing as a regularizer to implicitly enforce similar structural embeddings among neighboring nodes. This avoids explicitly calculating the costly matrix $\bar{\boldsymbol{A}}$, thus significantly reducing communication overhead and privacy risks.

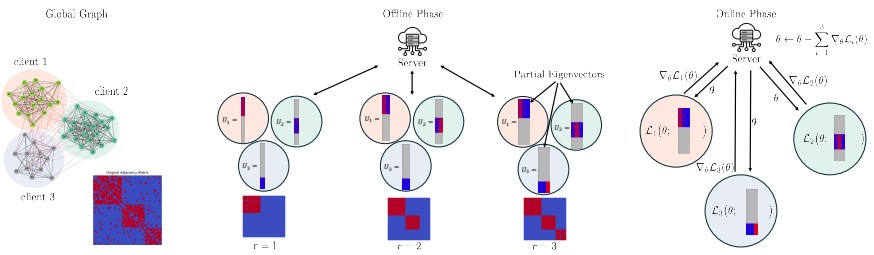

Figure 1: FEDLAP+ with three clients. Left: the global graph and its partitioning across clients. Center: local refinement of the global eigenvectors obtained via Arnoldi iterations; the corresponding adjacency matrix is shown below. Right: federated learning leveraging the estimated global eigenvectors.

- **FEDLAP+ (Section 4.2)** addresses the challenge posed by the large structural matrix $S$. It decomposes $S$ into a fixed spectral matrix $U \in \mathbb{R}^{n \times r}$ and a smaller learnable matrix $W \in \mathbb{R}^{r \times d_s}$. Instead of sharing the entire matrix $U$, FEDLAP+ distributes only the relevant rows to corresponding nodes. This efficient distribution is enabled by the spectral representation of the graph Laplacian, which allows truncation to retain only the smoothest eigenvectors. Consequently, this substantially reduces the dimensionality, accelerates convergence, and enhances privacy.

To efficiently compute the partial spectral decomposition in FEDLAP+, we **propose a decentralized version of the Arnoldi iteration (Section 4.3) and Appendix B.2**. This approach significantly reduces the computational cost of FEDLAP+, making it more efficient than prior frameworks (Section 6), scaling to large, sparse graphs, while preserving privacy (Section 5).

Below, we provide a detailed description and a formal analysis of these components.

## 4.1 FEDLAP: Exploiting Structural Information in SFL via Laplacian Smoothing

The core idea of FEDLAP is to leverage structural information through Laplacian smoothing, achieved by incorporating a graph Laplacian regularization term into the loss function in (1). Specifically, at each client $i \in [K]$, node prediction is performed for a node $v \in \mathcal{V}_i$ as

$$\hat{\boldsymbol{y}}_v = \text{softmax}\left(f_{\boldsymbol{\theta}_\mathsf{f}}(\boldsymbol{X}_i, \mathcal{E}_i, v) + g_{\boldsymbol{\theta}_\mathsf{s}}(\boldsymbol{s}_v)\right) , \qquad (2)$$

where the parameters of the model $\boldsymbol{\theta} = (\boldsymbol{\theta}_\mathsf{f}, \boldsymbol{\theta}_\mathsf{s}, \boldsymbol{S})$ are optimized based on the loss function

$$\mathcal{L}(\boldsymbol{\theta}) = \mathcal{L}_\mathsf{c}(\boldsymbol{\theta}) + \lambda_\text{reg} \frac{\text{Tr}(\boldsymbol{S}^\mathsf{T} \boldsymbol{L}_\mathcal{G} \boldsymbol{S})}{\text{Tr}(\boldsymbol{S}^\mathsf{T} \boldsymbol{S})} , \qquad (3)$$

with $\boldsymbol{L}_\mathcal{G}$ being the Laplacian matrix of graph $\mathcal{G}$, and $\boldsymbol{S}$ is generated using HOP2VEC [13] (see Appendix A).

In (3), the Laplacian regularizer is formulated using the Rayleigh Quotient, which normalizes the Laplacian term by the norm of $\boldsymbol{S}$. This normalization prevents the undesirable trivial minimization of the regularization term by simply reducing the norm of $\boldsymbol{S}$. Equation (3) can be rewritten as

$$\mathcal{L}(\boldsymbol{\theta}) = \mathcal{L}_\mathsf{c}(\boldsymbol{\theta}) + \lambda_\text{reg} \frac{\sum_{(u,v) \in \mathcal{E}} \|\boldsymbol{s}_u - \boldsymbol{s}_v\|^2}{\sum_{v \in \mathcal{V}} \|\boldsymbol{s}_v\|^2} . \qquad (4)$$

The regularization term is non-negative and decreases when neighboring nodes have similar NSFs.

The regularization term (3)–(4) implicitly captures pairwise relationships between nodes without clients necessitating the knowledge of the whole NSF matrix $\boldsymbol{S}$ and the local partition of $\bar{\boldsymbol{A}}$, as opposed to FEDSTRUCT. Specifically, Equation (4) shows that the Laplacian regularizer can be computed in a decentralized manner, where each client $i$ only requires the NSFs of its internal nodes and external neighbors, i.e., $\{\boldsymbol{s}_v, \quad \forall v \in \mathcal{V}_i \cup \mathcal{V}_i^*\}$. This approach not only enhances privacy compared to FEDSTRUCT but also significantly reduces communication overhead.

**Motivation.** Our motivation for employing Laplacian smoothing in FEDLAP arises from two critical considerations: (i) direct message passing in traditional SFL inherently risks exposing sensitive node and adjacency information, leading to privacy concerns; and (ii) graph convolutional networks (GCNs), as shown by Kipf and Welling [21], approximate spectral Laplacian smoothing through message passing. Hence, adopting Laplacian smoothing enables FEDLAP to implicitly leverage structural information without explicitly exchanging sensitive data, thus preserving the benefits of message-passing methods while addressing their privacy vulnerabilities in FL contexts.

Sharing NSFs from external nodes $\boldsymbol{s}_v \in \mathcal{V}_i^*$ may still pose privacy risks, as these features are indirectly tied to the labels through (2). Moreover, the high dimensionality of $\boldsymbol{S}$ makes its optimization computationally and communication-intensive, requiring multiple rounds of training, which amplifies the risk of information leakage.

To address these challenges and further enhance privacy, in Section 4.2 we propose leveraging the Laplacian regularizer in the spectral domain, as detailed in the next subsection. This approach eliminates the need for explicitly sharing $\boldsymbol{s}_v \in \mathcal{V}_i^*$.

## 4.2 FEDLAP+: Exploiting Structural Information in the Spectral Domain

FEDLAP+ is a spectral-domain variant of FEDLAP, designed to reduce communication overhead and privacy leakage while maintaining competitive performance. It decomposes the SFL problem into two distinct phases:

- An **offline phase** consisting of a one-time preprocessing step that precomputes the influence of the global graph structure for each node. This phase involves no model training and privately extracts useful graph-level structural information without revealing node features or labels.
- An **online (training) phase** that does not involve any exchange of information among clients and effectively reduces to standard FL.

The graph Laplacian $\boldsymbol{L}_{\mathcal{G}}$ is symmetric and positive semi-definite and can be decomposed as

$$\boldsymbol{L}_{\mathcal{G}} = \boldsymbol{U}\boldsymbol{\Lambda}\boldsymbol{U}^{\mathsf{T}}, \tag{5}$$

where $\boldsymbol{U} \in \mathbb{R}^{n \times n} = \begin{bmatrix} \boldsymbol{u}_1, \ldots, \boldsymbol{u}_n \end{bmatrix}$ is the matrix of orthonormal eigenvectors of $\boldsymbol{L}_{\mathcal{G}}$ and $\boldsymbol{\Lambda}$ is the diagonal matrix of eigenvalues, $\boldsymbol{\Lambda}_{j,j} = \lambda_j$, with $\lambda_1 \leq \cdots \leq \lambda_n$. Let $\boldsymbol{W} = \boldsymbol{U}^{\mathsf{T}}\boldsymbol{S} \in \mathbb{R}^{n \times d_{\mathsf{s}}}$ be the spectral representation of matrix $\boldsymbol{S}$. Substituting (5) into (2) and (3) yields

$$\hat{\boldsymbol{y}}_v = \mathrm{softmax}\left( f_{\boldsymbol{\theta}_{\mathsf{f}}}(\boldsymbol{X}_i, \mathcal{E}_i, v) + g_{\boldsymbol{\theta}_{\mathsf{s}}}(\boldsymbol{U}_{v,:}\boldsymbol{W}) \right) \tag{6}$$

$$\mathcal{L}(\boldsymbol{\theta}) = \mathcal{L}_{\mathsf{c}}(\boldsymbol{\theta}) + \lambda_{\mathrm{reg}} \frac{\mathrm{Tr}(\boldsymbol{W}^{\mathsf{T}}\boldsymbol{\Lambda}\boldsymbol{W})}{\mathrm{Tr}(\boldsymbol{W}^{\mathsf{T}}\boldsymbol{W})}. \tag{7}$$

where $\boldsymbol{U}_{v,:}$ is the $v$-th row of $\boldsymbol{U}$ and $\boldsymbol{\theta} = (\boldsymbol{\theta}_{\mathsf{f}}, \boldsymbol{\theta}_{\mathsf{s}}, \boldsymbol{W})$.

Leveraging the Laplacian in the spectral domain provides a principled way to truncate $\boldsymbol{W}$ and mitigate information exchange. In particular, since $\boldsymbol{\Lambda}$ is a diagonal matrix, we can simplify (7) as

$$\mathcal{L}(\boldsymbol{\theta}) = \mathcal{L}_{\mathsf{c}}(\boldsymbol{\theta}) + \lambda_{\mathrm{reg}} \frac{\sum_{j=1}^{n} \lambda_j \|\boldsymbol{w}_j\|^2}{\sum_{j=1}^{n} \|\boldsymbol{w}_j\|^2}, \tag{8}$$

where $\boldsymbol{w}_j$ is the $j$-th row of $\boldsymbol{W}$.

Equation (8) reveals that the Laplacian regularization term (3)–(4) acts as a *low-pass filter* by attenuating high-frequency components while preserving low-frequency (smooth) components of the graph signal. Specifically, minimizing (8) naturally reduces the coefficients $\|\boldsymbol{w}_j\|$ associated with high-frequency eigenvectors $\boldsymbol{u}_j$, which correspond to larger eigenvalues $\lambda_j$ of the graph Laplacian. This encourages the learned embeddings to align with low-frequency eigenvectors, which capture smooth variations across the graph. These eigenvectors correspond to signals that vary gradually across connected nodes, reflecting regions of high connectivity and structural continuity. As a result, the Laplacian regularization inherently promotes smoothness in the learned embeddings. This observation motivates truncating $\boldsymbol{W}$ by removing rows corresponding to large eigenvalues, as these represent less smooth—and consequently less informative—aspects of the graph structure.

To focus on the most informative spectral components and reduce dimensionality, we retain only the first $r \ll n$ rows of $\boldsymbol{W}$, defined as

$$\boldsymbol{W}_{[r],:} = \begin{bmatrix} \boldsymbol{w}_1^{\mathsf{T}}, \ldots, \boldsymbol{w}_r^{\mathsf{T}} \end{bmatrix}^{\mathsf{T}} \quad \in \mathbb{R}^{r \times d_{\mathsf{s}}}. \tag{9}$$

Similarly, we truncate the corresponding columns of $\boldsymbol{U}$ and the diagonal elements of $\boldsymbol{\Lambda}$:

$$\boldsymbol{U}_{:,[r]} = \begin{bmatrix} \boldsymbol{u}_1, \boldsymbol{u}_2, \ldots, \boldsymbol{u}_r \end{bmatrix} \quad \in \mathbb{R}^{n \times r}, \quad \boldsymbol{\Lambda}_{[r],[r]} = \mathrm{diag}(\lambda_1, \ldots, \lambda_r) \quad \in \mathbb{R}^{r \times r}. \tag{10}$$

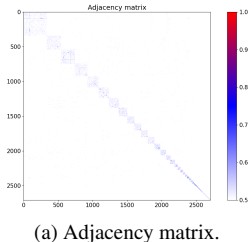 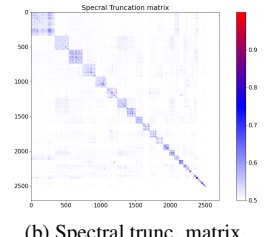 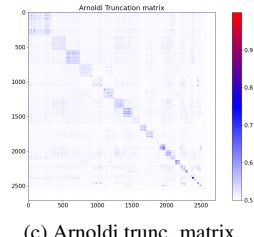

| (a) Adjacency matrix. | (b) Spectral trunc. matrix. | (c) Arnoldi trunc. matrix. |

Figure 2: Comparison of different matrix representations in the graph. In (b) and (c), $r = 100$ for dimensionality reduction.

With this truncation, the graph Laplacian $\boldsymbol{L}_{\mathcal{G}}$ (see (5)) can be approximated as

$$\boldsymbol{L}_{\mathcal{G}} \approx \boldsymbol{U}_{:,[r]} \boldsymbol{\Lambda}_{[r],[r]} \boldsymbol{U}_{:,[r]}^{\mathsf{T}} . \tag{11}$$

To obtain a good approximation of the Laplacian $\boldsymbol{L}_{\mathcal{G}}$, $r$ should be chosen on the order of its rank, i.e., the number of communities in $\mathcal{G}$, which is much smaller than $n$. In Fig. 2, we apply spectral truncation to the Cora dataset (2708 nodes) and compare the reconstructed adjacency matrix (Fig.2(b)) with the original (Fig. 2(a)). As shown, the global structure is preserved, yielding a smoother, low-pass version of the graph.

The truncation of spectral components not only drastically reduces communication overhead (thereby improving privacy), but also serves as an additional form of regularization, preventing the model from overfitting to noise or irrelevant details in the graph structure. This is particularly important in FL settings, where models must generalize well across different subgraphs from multiple clients.

Note that FEDLAP+ inherits the standard convergence guarantees of FEDAVG (see Appendix D).

### 4.3 Decentralized Arnoldi Iteration: Privacy-Preserving Approximation of the Laplacian

FEDLAP+ requires the eigendecomposition of the Laplacian $\boldsymbol{L}_{\mathcal{G}}$. While a full decomposition has a complexity of $\mathcal{O}(n^3)$ and is prohibitively expensive in decentralized settings, as discussed in Section 4.2, FEDLAP+ only requires the first $r$ eigenvectors associated with the smallest eigenvalues. To compute these efficiently and in a privacy-preserving manner, we propose a decentralized version of the Arnoldi iteration [23], particularly well-suited for large, sparse graphs. As detailed in Appendix B.3, its **complexity is $\mathcal{O}(nr^2)$, i.e., linear in $n$**, under typical sparsity assumptions.

The Arnoldi iteration is an efficient iterative method for approximating eigenvalues and eigenvectors of large, sparse matrices. Rather than performing a full (and potentially very costly) eigendecomposition, Arnoldi constructs an orthonormal basis for the so-called Krylov subspace $\mathcal{K}_m(\boldsymbol{M}, \boldsymbol{x}) = \mathrm{span}\{\boldsymbol{x}, \boldsymbol{M}\boldsymbol{x}, \ldots, \boldsymbol{M}^{m-1}\boldsymbol{x}\}$, where $\boldsymbol{x}$ is some chosen starting vector. Specifically, it computes an orthonormal basis $\{\boldsymbol{q}_1, \ldots, \boldsymbol{q}_m\}$ for the subspace $\mathcal{K}_m(\boldsymbol{M}, \boldsymbol{x})$ iteratively and yields an approximate eigendecomposition of $\boldsymbol{M}$ as

$$\boldsymbol{M} \approx \boldsymbol{U}\boldsymbol{\Sigma}\boldsymbol{U}^{\mathsf{T}}, \tag{12}$$

where $\boldsymbol{U} = \boldsymbol{Q}_m \boldsymbol{V}$ and $\boldsymbol{U}^{\mathsf{T}}\boldsymbol{U} \approx \boldsymbol{I}$, with $\boldsymbol{Q}_m = [\boldsymbol{q}_1, \ldots, \boldsymbol{q}_m]$ being the matrix of Arnoldi basis vectors, and $\boldsymbol{V}$ and $\boldsymbol{\Sigma}$ the matrix of eigenvectors and eigenvalues, respectively, of an upper Hessenberg matrix $\boldsymbol{H}_m \in \mathbb{R}^{m \times m}$ with entries $h_{ij} = \boldsymbol{q}_i^{\mathsf{T}} \boldsymbol{M} \boldsymbol{q}_j$. For details, we refer the reader to Appendix B.

We use the Arnoldi iteration to approximate the eigenvalues and eigenvectors of $\boldsymbol{L}_{\mathcal{G}}$. Crucially, the Arnoldi iteration relies only on matrix-vector multiplication. As shown in Section 5, this enables a decentralized, privacy-preserving implementation that does not disclose clients' node structures. In particular, given the Krylov subspace $\mathcal{K}_m(\boldsymbol{L}_{\mathcal{G}}, \boldsymbol{v})$, the Arnoldi update becomes (see (22) in Appendix B)

$$\boldsymbol{r}_\ell = \boldsymbol{L}_{\mathcal{G}} \boldsymbol{q}_\ell - \sum_{i=1}^{\ell} h_{i,\ell} \boldsymbol{q}_i, \quad h_{i,\ell} = \boldsymbol{q}_i^{\mathsf{T}} \boldsymbol{L}_{\mathcal{G}} \boldsymbol{q}_\ell, \quad \boldsymbol{q}_{\ell+1} = \frac{\boldsymbol{r}_\ell}{\|\boldsymbol{r}_\ell\|} . \tag{13}$$

More compactly, if we stack the first $r$ Arnoldi vectors in $\boldsymbol{Q}_r = [\boldsymbol{q}_1, \ldots, \boldsymbol{q}_r]$ and let $\boldsymbol{H}_r \in \mathbb{R}^{r \times r}$ collect the coefficients $h_{ij} = \boldsymbol{q}_i^{\top} \boldsymbol{L}_{\mathcal{G}} \boldsymbol{q}_j$, we obtain the Arnoldi relation

$$\boldsymbol{L}_{\mathcal{G}} \boldsymbol{Q}_r = \boldsymbol{Q}_r \boldsymbol{H}_r + h_{r+1,r} \boldsymbol{q}_{r+1} \boldsymbol{e}_r^{\top} , \tag{14}$$

where $\boldsymbol{e}_r$ is the $r$-th standard basis vector. A small residual $h_{r+1,r}$ implies the rank-$r$ approximation

$$\boldsymbol{L}_{\mathcal{G}} \approx \boldsymbol{Q}_r \boldsymbol{H}_r \boldsymbol{Q}_r^\top = \boldsymbol{Q}_r \boldsymbol{V}_r \boldsymbol{\Sigma}_r \boldsymbol{V}_r^\top \boldsymbol{Q}_r^\top, \tag{15}$$

where $\boldsymbol{V}_r \boldsymbol{\Sigma}_r \boldsymbol{V}_r^\top$ is the eigendecomposition of $\boldsymbol{H}_r$. Defining $\boldsymbol{U}_{:,[r]} \triangleq \boldsymbol{Q}_r \boldsymbol{V}_r$ and $\boldsymbol{\Lambda}_{[r],[r]} \triangleq \boldsymbol{\Sigma}_r$ recovers the truncated Laplacian approximation in (11).

**Proposed decentralized Arnoldi iteration.** We aim to use Arnoldi to estimate the smallest $r$ eigenvalues of $\boldsymbol{L}_{\mathcal{G}}$ and corresponding eigenvectors in a decentralized manner across clients while preserving privacy. For a generic vector $\boldsymbol{q}$, we define $\boldsymbol{b} = \boldsymbol{L}_{\mathcal{G}} \boldsymbol{q}$. As each client $i$ knows the rows and columns of the adjacency matrix indexed by $\mathcal{V}_i$, i.e., $\boldsymbol{A}_{\mathcal{V}_i,:}$ and $\boldsymbol{A}_{:,\mathcal{V}_i}$ (and thus also $\boldsymbol{D}_{\mathcal{V}_i,:}$ and $\boldsymbol{D}_{:,\mathcal{V}_i}$), client $i$ needs to obtain its local block of $\boldsymbol{b} = \boldsymbol{L}_{\mathcal{G}} \boldsymbol{q}$, namely $\boldsymbol{b}_{\mathcal{V}_i}$. This block can be written as

$$\boldsymbol{b}_{\mathcal{V}_i} = \boldsymbol{D}_{\mathcal{V}_i,\mathcal{V}_i} \boldsymbol{q}_{\mathcal{V}_i} - \sum_{j=1}^{K} \boldsymbol{A}_{\mathcal{V}_i,\mathcal{V}_j} \boldsymbol{q}_{\mathcal{V}_j}. \tag{16}$$

The first term is computable using only local information, whereas the second term requires collaboration across clients. To preserve privacy, so that no party learns any part of the global adjacency beyond its own, each client $j$ computes the local product $\boldsymbol{A}_{\mathcal{V}_i,\mathcal{V}_j} \boldsymbol{q}_{\mathcal{V}_j}$ and sends an *additively homomorphically* encrypted ciphertext to the server. The server sums these ciphertexts over all $j \in [K]$ and returns the encrypted aggregate to client $i$, who decrypts it to obtain $\sum_{j=1}^{K} \boldsymbol{A}_{\mathcal{V}_i,\mathcal{V}_j} \boldsymbol{q}_{\mathcal{V}_j}$. In this way, the server never accesses individual contributions in plaintext, and client $i$ learns only the required sum. Protocol details are in Appendix B.2, and the privacy analysis is given in Section 5 and Appendix C.

## 5 Privacy Analysis of FEDLAP+

In this section, we analyze the privacy of FEDLAP+. We show that, under a strong attacker model, clients cannot infer other clients' internal connections or cross-client connections, i.e., FEDLAP+ provides strong privacy.

As mentioned earlier, FEDLAP+ is divided into an **offline** and an **online** phase. In the **online phase**, clients federate the model parameters $\boldsymbol{\theta} = (\boldsymbol{\theta}_{\mathsf{f}}, \boldsymbol{\theta}_{\mathsf{s}}, \boldsymbol{W})$ via an arbitrary FL scheme, e.g., FEDAVG [4]. Hence, the online phase of FEDLAP+ exhibits the same kind of vulnerabilities as FL and is amenable to privacy enhancing techniques like differential privacy, homomorphic encryption, and secure aggregation [24]. In the **offline phase**, executed once before training, no node features or labels are shared; only information related to the graph structure is exchanged. The goal is to extract a compact structural summary while preserving privacy. As previously explained, we operate in the spectral domain and use a decentralized Arnoldi procedure to estimate a small set of Laplacian eigenvectors, leveraging the empirical fact that most interconnection signals lie in low-frequency components.

Under this decomposition, any *additional* privacy considerations specific to FEDLAP+ are confined to the *offline* phase, as the online phase introduces no leakage beyond standard FL. In the offline phase, since no features or labels leave a client, the only potential leakage channel pertains to *edges*. We thus focus on structural privacy and cast the attack as a **membership-inference attack** on edges: given the offline messages, can an adversary determine whether a specific connection $A_{uv}$ "participated" in the Arnoldi computations? This results in a binary hypothesis test based on a log-likelihood ratio (LLR). By the Neyman-Pearson lemma, the LLR test is the optimal decision rule for this setting.

**Attacker observations and procedure.** For the analysis, we consider a worst-case scenario involving two clients: client 1 (target) and client 2 (attacker). The attacker aims to infer whether an edge exists between two nodes $u, v \in \mathcal{V}_1$ (test $H_0 : A_{uv} = 0$ vs. $H_1 : A_{uv} = 1$). From the decentralized Arnoldi updates (see (16)) the attacker obtains, for the target client, the aggregated vector $\boldsymbol{\tau}_{\mathcal{V}_2} = \boldsymbol{A}_{\mathcal{V}_2,\mathcal{V}_1} \boldsymbol{q}_{\mathcal{V}_1}$, and also knows the adjacency blocks $\boldsymbol{A}_{\mathcal{V}_2,\mathcal{V}_1}$. Since $\boldsymbol{\tau}_{\mathcal{V}_2}$ comprises $n_2$ linear equations in the $n - n_2$ unknown spectral blocks $\boldsymbol{q}_{\mathcal{V}_1}$, the attacker can only form an *estimate* $\check{\boldsymbol{q}}_{\mathcal{V}_1}$ of the true spectral basis. We assume $\|\check{\boldsymbol{Q}} - \boldsymbol{Q}_{\mathcal{V}_1,:}\| \leq \sigma$, where $\check{\boldsymbol{Q}}$ is the estimate of $\boldsymbol{Q}_{\mathcal{V}_1,:}$ and $\sigma$ quantifies the attacker's uncertainty. Using the public $\boldsymbol{H}_r$ and its spectral estimate $\check{\boldsymbol{Q}}$, and invoking the Arnoldi relation (14), the attacker creates the equation

$$\boldsymbol{U} \approx \check{\boldsymbol{A}} \check{\boldsymbol{Q}}, \tag{17}$$

where $\boldsymbol{U} \triangleq \boldsymbol{D}_{\mathcal{V}_1,\mathcal{V}_1} \check{\boldsymbol{Q}} + \boldsymbol{A}_{\mathcal{V}_1,\mathcal{V}_2} \boldsymbol{Q}_{\mathcal{V}_2,:} - \check{\boldsymbol{Q}} \boldsymbol{H}_r$ and $\check{\boldsymbol{A}} = \boldsymbol{A}_{\mathcal{V}_1,\mathcal{V}_1}$. Equality in (17) holds only when $\sigma = 0$ and $h_{r+1,r} = 0$. The attacker must also know $\boldsymbol{D}_{\mathcal{V}_1,\mathcal{V}_1}$ to calculate $\boldsymbol{U}$. The attacker then

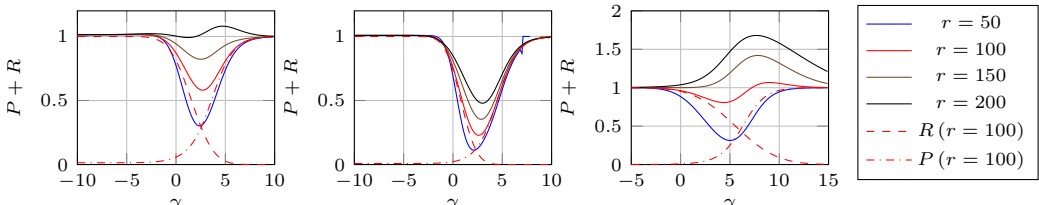

Figure 3: Effect of the rank parameter $r$ on the precision + recall with varying $\gamma$ for the Chameleon (left), Amazon photo (center), and PubMed (right) datasets. The pairs $(p, n)$ are $(0.0139, 2277)$, $(0.008, 7650)$, and $(0.0005, 19717)$, respectively. The curves illustrate how the choice of $r$ impacts the trade-off between recall and precision as the decision threshold varies.

performs the following steps: (i) obtain $U$, (ii) evaluate the log-likelihood ratio $\text{LLR}_{u,v}$ for the two hypotheses using $U$, and (iii) decide $H_1$ whenever $\text{LLR}_{u,v} \geq \gamma$ for some threshold $\gamma \in \mathbb{R}$.

We note that the following analysis adopts a deliberately conservative perspective. In line with standard practice in privacy research (e.g., secure aggregation and spectral privacy frameworks), we assume an unrealistically strong attacker to obtain a worst-case privacy guarantee.

**Theorem 1.** *Consider two clients running the decentralized Arnoldi scheme outlined in Sec. 4.3. Let $A$ be a random graph with $p$ denoting the probability of a connection between any pair $(u, v)$ for $u, v \in \mathcal{V}_1$. Assume $p$ to be known by client 2. Let $U = \breve{A}\breve{Q}$, where $\breve{A} = A_{\mathcal{V}_1, \mathcal{V}_1}$ and $\breve{Q} \approx Q_{\mathcal{V}_1,:}$ are client 2's observations (provided by client 1) about the sensitive low-rank matrix $\breve{A}$. Moreover, let $\breve{Q}$ have delocalized entries and be known to client 2. For large $n$, the LLR*

$$\text{LLR}_{u,v} = \log\left(\frac{P(U|\breve{A}_{uv} = 1)}{P(U|\breve{A}_{uv} = 0)}\right), \tag{18}$$

*is a random variable with the distribution*

$$H_1: \quad \text{LLR}_{u,v} \sim \mathcal{N}\left(\frac{1}{2}\alpha_v, \alpha_v\right), \quad H_0: \quad \text{LLR}_{u,v} \sim \mathcal{N}\left(-\frac{1}{2}\alpha_v, \alpha_v\right) \tag{19}$$

*where $\alpha_v = \breve{Q}_{v,:}\Sigma^{-1}\breve{Q}_{v,:}^{\mathsf{T}}$ and $\Sigma = p(1-p)\breve{Q}^{\mathsf{T}}\breve{Q}$.*

*Proof.* See Appendix C.3. $\qquad\square$

Theorem 1 provides insights into how different parameters influence the attack performance, as shown in Corollary 1.

**Corollary 1.** *Consider the same setting as in Theorem 1. If $\breve{Q}^{\mathsf{T}}\breve{Q} \approx (r/n)I_r$ it follows that*

$$D_{\text{KL}}\left(\Pr(\text{LLR}_{u,v} \mid H_1) \,\|\, \Pr(\text{LLR}_{u,v} \mid H_0)\right) \approx \frac{r}{2np(1-p)}.$$

*Proof.* See Appendix C.4 $\qquad\square$

The KL divergence in Corollary 1 quantifies the discrepancy of the LLR distributions under the two hypotheses; a lower value makes it harder for the adversary to distinguish between them. This implies that a larger $r$, i.e., increased shared information between clients, and a smaller $p$, i.e., sparser graphs, negatively impact privacy, whereas a greater number of nodes in the graph, $n$, has a beneficial effect.

Using Theorem 1, in Appendix C.5 we derive the true-positive rate (TPR) and the false-positive rate (FPR) for the attack and then use them to obtain expressions for the precision and recall of the attack as a function of $p$, $r$, $n$, and $\gamma$. In Fig. 3, we plot the sum of precision ($P$) and recall ($R$) for different values of $r$ and varying $\gamma$. Each figure corresponds to a distinct pair $(p, n)$ drawn from three different datasets—Chameleon, Amazon photo, and PubMed—where $p$ is the estimated probability of a connection and $n$ is the number of nodes. We observe that, for sufficiently small $r$, $P + R \leq 1$ ($r = 175$ for Chameleon, $r = 350$ for Amazon-Photo, and $r = 80$ for PubMed). In our privacy analysis, achieving $P + R < 1$ indicates that the attacker gains no meaningful advantage over trivial assumptions—either all nodes connected (precision $P \approx 0$, recall $R = 1$) or all disconnected—thus revealing no useful information about individual connections.

Table 1: Communication cost.

| Algorithm | Offline | Online |
|---|---|---|
| FEDLAP | 0 | $\mathcal{O}(E \cdot K \cdot |\boldsymbol{\theta}| + E \cdot K \cdot d \cdot n)$ |
| FEDLAP+ (Arnoldi) | $\mathcal{O}(r \cdot K \cdot n)$ | $\mathcal{O}(E \cdot K \cdot |\boldsymbol{\theta}|)$ |
| FEDSTRUCT | $\mathcal{O}(L_{\mathrm{s}} \cdot K \cdot p \cdot n)$ | $\mathcal{O}(E \cdot K \cdot |\boldsymbol{\theta}| + E \cdot K \cdot d \cdot n)$ |
| FEDGCN-2HOP | $\mathcal{O}(n \cdot d \cdot c_{\mathrm{avg}})$ | $\mathcal{O}(E \cdot K \cdot |\boldsymbol{\theta}|)$ |
| FEDSAGE+ | 0 | $\mathcal{O}(E \cdot K^2 \cdot |\boldsymbol{\theta}| + E \cdot K \cdot d \cdot n)$ |
| FEDAVG | 0 | $\mathcal{O}(E \cdot K \cdot |\boldsymbol{\theta}|)$ |

Figure 4: Precision vs recall on PubMed.

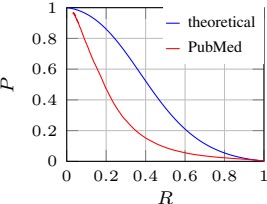

In Fig. 4, we compare the theoretical attack performance, derived in App.C.5, with an actual attack on the links in PubMed. The theoretical results are based on several assumptions (see Thm. 1) that do not apply to the real attack. As shown in the figure, the actual attack is weaker than the theoretical predictions. Notably, these results assume an exceptionally strong, albeit unrealistic, attacker with knowledge of $\check{Q}$ and $p$.

**Remark.** No formal privacy analysis exists for FEDSTRUCT or FEDGCN, making FEDLAP especially appealing. Analyzing their privacy is challenging due to the iterative information exchange in the online phase. In Appendix C.6, we provide arguments supporting the stronger privacy of FEDLAP.

## 6 Communication Complexity

Table 1 displays the communication complexity of FEDLAP+ alongside other SFL schemes. The communication complexity is divided into two parts: pre-training, a setup phase to acquire components necessary for training, and an online phase where the actual training takes place. In the table, $E$, $K$, and $n$ represent the number of training rounds, clients, and nodes in the graph, respectively. Moreover, for simplicity, we assume all feature dimensions to be equal to $d$ and $|\boldsymbol{\theta}|$ to be the model size. $L_{\mathrm{s}}$ and $p$ are the number of layers and pruning parameter of FEDSTRUCT and, to incorporate FEDGCN, we consider the average number of clients that contain neighbors to a given node, $c_{\mathrm{ave}}$. As seen in the table, the online complexity of FEDLAP+ is on par with FEDAVG and FEDGCN but is significantly lower than that of FEDSAGE+ and FEDSTRUCT. In the pre-training phase, FEDLAP+ scales with $n$, as does FEDSTRUCT and FEDGCN (which does not provide privacy). However, the other parameters are typically much smaller in FEDLAP+ than in its counterparts. Particularly, FEDGCN suffers for large $c_{\mathrm{avg}}$ and $d$, as can be seen in Appendix F.

## 7 Experimental Results

In this section, we evaluate the performance of FEDLAP on node classification for varying client counts, and limited number of training nodes (only $10\%$ of the total nodes). We report results alongside the edge homophily ratio $h \in [0, 1]$, which quantifies the proportion of edges connecting nodes with the same label [25]. Experiments are conducted on six datasets: Cora and Citeseer [26], PubMed [27], Chameleon [28], Amazon Photo [29], and Ogbn-Arxiv [30]. Our experiments were conducted on a machine with $2 \times$ NVIDIA Tesla V100 SXM2 GPUs, each with 32GB of RAM.

To assess robustness across different partitioning strategies, we provide results using **Louvain** and **KMeans** partitionings in Appendix E.1. The centralized and local settings constitute upper and lower bounds on the performance and are included for reference. We also incorporate several benchmark methods including FEDSAGE+ [5], FEDPUB [11], FEDGCN [10], and FEDSTRUCT [13]. Each of these methods share sensitive information that may violate privacy of the node features or links. On the contrary, both FEDLAP and FEDLAP+ significantly reduce the amount of shared information and does not leak information even under very severe attack threats as shown in Section 5.

**Performance Analysis.** In Table 2, we report the average accuracy over 10 runs across six datasets with random partitioning. FEDLAP and FEDLAP+ outperform general FL baselines like FEDSGD, FEDSAGE+, and FEDPUB, and remain competitive with structure-aware methods such as FEDGCN and FEDSTRUCT—while providing stronger privacy guarantees. Notably, FEDGCN requires 2-hop aggregation sharing (see Appendix C4 in [13] for a discussion), and FEDSTRUCT involves iterative sharing of structural features, both leading to potential privacy leakage.

FEDLAP excels on homophilic graphs (e.g., Pubmed, Cora) where Laplacian smoothing is effective, but underperforms on heterophilic graphs like Chameleon, where neighboring nodes behave differently. FEDLAP+, by contrast, remains robust across all datasets by operating in the spectral

Table 2: Node classification accuracy with random partitioning. Nodes are split into train-val-test as 10%-10%-80%. For each result, the mean and standard deviation are shown for 10 independent runs. Edge homophily ratio ($h$) is given in brackets.

| | CORA ($h = 0.81$) | | | CITESEER ($h = 0.74$) | | | PUBMED ($h = 0.80$) | | |
|---|---|---|---|---|---|---|---|---|---|
| CENTRAL GNN | 83.40± 0.63 | | | 70.99± 0.32 | | | 85.60± 0.26 | | |
| | 5 CLIENTS | 10 CLIENTS | 20 CLIENTS | 5 CLIENTS | 10 CLIENTS | 20 CLIENTS | 5 CLIENTS | 10 CLIENTS | 20 CLIENTS |
| FEDSGD GNN | 65.46± 2.45 | 65.26± 1.37 | 64.38± 1.38 | 66.84± 1.02 | 66.53± 1.03 | 66.11± 1.11 | 84.24± 0.29 | 83.96± 0.19 | 83.56± 0.27 |
| FEDSAGE+ | 65.80± 1.72 | 64.53± 1.54 | 63.62± 1.08 | 66.64± 0.98 | 66.57± 0.67 | 66.24± 0.89 | 84.29± 0.37 | 83.96± 0.23 | 83.55± 0.27 |
| FEDPUB | 68.22± 1.10 | 59.17± 1.34 | 47.91± 1.98 | 64.86± 0.97 | 63.30± 1.82 | 56.00± 2.22 | 84.13± 0.19 | 84.00± 0.21 | 83.45± 0.22 |
| FEDGCN-2HOP | 81.48± 0.81 | 82.22± 0.79 | 82.82± 0.73 | 71.36± 0.60 | 71.75± 0.80 | 69.71± 0.54 | 85.93± 0.29 | 86.13± 0.34 | 85.90± 0.28 |
| FEDSTRUCT-P (H2V) | 79.02± 0.93 | 80.01± 1.00 | 80.09± 0.60 | 67.71± 0.96 | 67.51± 1.01 | 64.54± 1.62 | 85.41± 0.21 | 85.40± 0.17 | 85.27± 0.25 |
| FEDLAP | 80.85± 1.24 | 80.55± 0.97 | 80.42± 0.69 | 67.24± 0.91 | 66.29± 0.85 | 63.96± 1.66 | 86.27± 0.31 | 86.43± 0.19 | 85.86± 0.23 |
| FEDLAP+ (ARNOLDI) | 79.57± 1.00 | 79.31± 1.03 | 79.42± 1.23 | 67.80± 0.98 | 67.20± 0.98 | 65.52± 1.65 | 85.22± 0.33 | 85.29± 0.26 | 85.05± 0.38 |
| LOCAL GNN | 47.48± 1.85 | 37.59± 1.12 | 32.66± 1.20 | 51.93± 0.64 | 49.94± 1.66 | 40.33± 1.20 | 33.23± 0.7 | 76.77± 0.25 | 72.59± 0.41 |

| | CHAMELEON ($h = 0.23$) | | | AMAZON PHOTO ($h = 0.82$) | | | OGBN-ARXIV ($h = 0.65$) | | |
|---|---|---|---|---|---|---|---|---|---|
| CENTRAL GNN | 54.38± 1.60 | | | 94.07± 0.41 | | | 68.04± 0.09 | | |
| | 5 CLIENTS | 10 CLIENTS | 20 CLIENTS | 5 CLIENTS | 10 CLIENTS | 20 CLIENTS | 5 CLIENTS | 10 CLIENTS | 20 CLIENTS |
| FEDSGD GNN | 40.97± 0.94 | 35.93± 1.62 | 34.41± 1.95 | 91.40± 0.41 | 89.93± 0.56 | 89.12± 0.59 | 57.10± 0.17 | 54.07± 0.10 | 51.74± 0.20 |
| FEDSAGE+ | 39.96± 1.17 | 35.15± 1.99 | 34.59± 2.31 | 91.46± 0.52 | 89.97± 0.58 | 89.15± 0.56 | ? | ? | ? |
| FEDPUB | 38.45± 2.17 | 34.24± 2.40 | 29.41± 2.44 | 89.73± 0.72 | 88.03± 0.76 | 85.48± 0.83 | 59.12± 0.13 | 55.50± 0.11 | 52.15± 0.12 |
| FEDGCN-2HOP | 51.51± 1.46 | 50.19± 1.34 | 52.04± 1.13 | 93.61± 0.28 | 93.36± 0.44 | 93.73± 0.40 | 66.77± 0.13 | 66.93± 0.14 | 66.89± 0.08 |
| FEDSTRUCT-P (H2V) | 55.65± 1.22 | 55.81± 1.69 | 55.78± 1.68 | 92.47± 0.35 | 92.00± 0.51 | 92.51± 0.27 | 65.17± 0.16 | 64.95± 0.06 | 64.94± 0.22 |
| FEDLAP | 32.91± 2.45 | 32.98± 2.63 | 32.85± 1.88 | 92.24± 0.44 | 92.08± 0.73 | 92.26± 0.36 | 66.60± 0.26 | 66.03± 0.33 | 65.93± 0.40 |
| FEDLAP+ (ARNOLDI) | 53.53± 1.33 | 54.34± 1.59 | 54.15± 0.91 | 92.59± 0.36 | 92.14± 0.56 | 92.79± 0.32 | 66.73± 0.15 | 66.22± 0.26 | 66.06± 0.26 |
| LOCAL GNN | 36.06± 1.53 | 36.06± 1.53 | 29.53± 1.54 | 24.93± 1.01 | 77.62± 0.84 | 60.97± 1.32 | 55.46± 0.16 | 50.43± 0.15 | 45.34± 0.14 |

[2]**FEDGCN lacks privacy** as the server must have access to aggregated node features and 2-hop structures are shared between clients, which constitutes a privacy breach as shown in [16]. Also, the official code overlooks isolated external neighbors removal, potentially enhancing prediction performance above its actual capabilities.

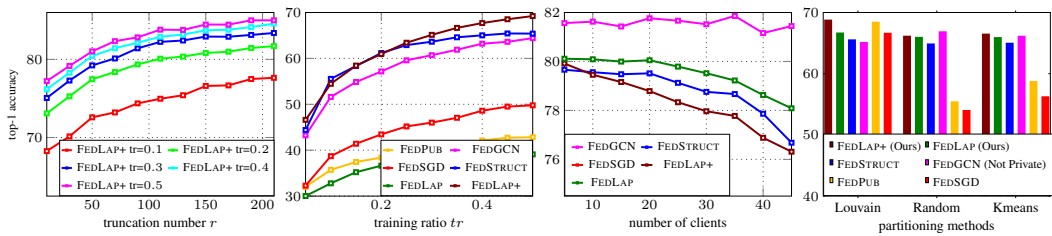

Figure 5: Left to right: (i) Accuracy vs $r$ for various training ratios ($tr$) on Cora (10 clients, random partitioning); (ii) Accuracy vs training ratio on Chameleon (Kmeans partitioning); (iii) Accuracy vs num of clients on Cora (random partitioning); (iv) Accuracy on OGBN-Arxiv (10 clients) on various partitioning methods.

domain and applying truncation, which filters noisy signals and avoids the limitations of smoothing. Though truncation reduces information, it regularizes learning and simplifies optimization, which helps FEDLAP+ perform well in low-label or large-scale settings (e.g., ogbn-arxiv). In summary, FEDLAP+ is more robust on heterophilic and large graphs, while FEDLAP favors high privacy and communication efficiency—justifying slight utility trade-offs in some cases.

Fig. 5 demonstrates strong and consistent performance for FEDLAP and FEDLAP+. Small truncation numbers ($r$) already yield high accuracy (left), showing that only a few dominant spectral components are sufficient to capture the global structure. Accuracy remains robust across training ratios (mid-left) and scales smoothly with the number of clients (mid-right), confirming that FEDLAP+ maintains stability even under highly partitioned data. On OGBN-Arxiv (right), both methods outperform all alternatives across different partitioning strategies, with FEDLAP+ particularly excelling on larger and more heterogeneous graphs. Note that, in practice, moderate values of $r$ (e.g., 50–200) provide an excellent balance between accuracy and efficiency, as increasing $r$ further offers only marginal gains. Additional experimental results are reported in Appendix E.

**Concluding remarks.** FEDLAP achieves performance close to the centralized setting and significantly outperforms prior methods such as FEDSAGE+ and FEDPUB in challenging settings. It also matches the performance of FEDSTRUCT and even the non-private FEDGCN, while being the first SFL method to provide strong privacy guarantees. By doing so, FEDLAP advances the Pareto frontier in the accuracy–privacy–communication space, demonstrating that strong privacy and low communication overhead can be attained without sacrificing accuracy. Although this paper focuses on node classification, the proposed framework is applicable to any local graph-based task, including edge prediction and link-level inference.

**Acknowledgments**

This work was partially supported by the Swedish Research Council (VR) under grants 2020-03687 and 2023-05065, by the Wallenberg AI, Autonomous Systems and Software Program (WASP) funded by the Knut and Alice Wallenberg Foundation, and by the Swedish innovation Agency (Vinnova) under grant 2022-03063.

The computations were enabled by resources provided by the National Academic Infrastructure for Supercomputing in Sweden (NAISS), partially funded by the Swedish Research Council through grant agreement no. 2022-06725.

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

# A  Prediction model with FEDSTRUCT

The proposed SFL scheme in Section 4 aligns with the philosophy of FEDSTRUCT [13] by utilizing explicit global graph structure information to enhance performance. However, it overcomes the limitations of FEDSTRUCT through a fundamentally different approach—integrating this information through the Laplacian. For clarity, we provide a concise overview of FEDSTRUCT below.

In FEDSTRUCT, the prediction for a node $v \in \mathcal{V}$ is given by

$$\hat{\boldsymbol{y}}_v = \text{softmax}\left(\boldsymbol{h}_v + \boldsymbol{z}_v\right), \tag{20}$$

where $\boldsymbol{h}_v$ is the *node feature embedding* (NFE) and $\boldsymbol{z}_v$ the *node structure embedding*, which encodes structural information of the node. The NFEs $\boldsymbol{h}_v$ are computed locally at each client by a GNN based on the local node features and local connections, $\boldsymbol{h}_v = f_{\boldsymbol{\theta}_\mathrm{f}}(\boldsymbol{X}_i, \mathcal{E}_i, v)$, where $\boldsymbol{\theta}_\mathrm{f}$ are the learnable parameters of the GNN.

The NSEs $\boldsymbol{z}_v$ are generated based on *node structure features* (NSFs), which encode structural properties of a node, such as degree and neighborhood patterns, providing a task-specific representation of the graph topology. Let $\boldsymbol{s}_v \in \mathbb{R}^{d_\mathrm{s}}$ be the NSF of node $v$ and $\boldsymbol{S}$ the matrix containing all NSFs as rows, $\boldsymbol{S} = [\boldsymbol{s}_1^\mathsf{T}, \ldots, \boldsymbol{s}_n^\mathsf{T}]^\mathsf{T}$. Particularly, the NSEs are computed as

$$\boldsymbol{z}_v = \sum_{u \in V} \bar{A}_{vu} g_{\boldsymbol{\theta}_\mathrm{s}}(\boldsymbol{s}_u), \tag{21}$$

where $g_{\boldsymbol{\theta}_\mathrm{s}}$ is a learnable function parameterized by $\boldsymbol{\theta}_\mathrm{s}$.

The NSFs $\boldsymbol{s}_u$ can be generated using established methods such as GDV or NODE2VEC. However, these approaches rely on knowledge of the global graph. To address this limitation, [13] proposed HOP2VEC, which generates task-dependent NSFs, without access to the global graph, by treating them as learnable features optimized dynamically during training.

# B  Details of the Arnoldi Iteration Method

## B.1  Standard Arnoldi Iteration

The Arnoldi iteration is an efficient iterative method for approximating eigenvalues and eigenvectors of large, sparse matrices. Rather than performing a full (and potentially very costly) eigendecomposition, Arnoldi constructs an orthonormal basis for the so-called Krylov subspace $\mathcal{K}_m(\boldsymbol{M}, \boldsymbol{x}) = \text{span}\{\boldsymbol{x}, \boldsymbol{M}\boldsymbol{x}, \ldots, \boldsymbol{M}^{m-1}\boldsymbol{x}\}$, where $\boldsymbol{x}$ is some chosen starting vector.

Given an orthonormal basis $\{\boldsymbol{q}_1, \ldots, \boldsymbol{q}_\ell\}$ for the subspace $\mathcal{K}_m(\boldsymbol{M}, \boldsymbol{v})$, the Arnoldi method iteratively computes the next basis vector $\boldsymbol{q}_{\ell+1}$ as

$$\boldsymbol{q}_{\ell+1} = \frac{\boldsymbol{r}_\ell}{\|\boldsymbol{r}_\ell\|}, \quad \boldsymbol{r}_\ell = \boldsymbol{M}\boldsymbol{q}_\ell - \sum_{i=1}^{\ell} h_{i,\ell}\boldsymbol{q}_i, \tag{22}$$

where $h_{i,\ell} = \boldsymbol{q}_i^\mathsf{T} \boldsymbol{M} \boldsymbol{q}_\ell$. The vectors $\boldsymbol{q}$ are also referred to as Arnoldi vectors.

From this orthonormal basis, the method constructs an approximate decomposition to estimate some of the eigenvalues and eigenvectors of $\boldsymbol{M}$.

Note that

$$\begin{aligned}
\|\boldsymbol{r}_m\| &= \boldsymbol{q}_{m+1}^\mathsf{T} \boldsymbol{r}_\ell \\
&= \boldsymbol{q}_{\ell+1}^\mathsf{T} \boldsymbol{M} \boldsymbol{q}_\ell \\
&= h_{\ell+1,\ell},
\end{aligned} \tag{23}$$

where the second equality follows since, by construction, $\boldsymbol{q}_{\ell+1}$ is orthogonal to all the previous Arnoldi vectors.

Using (23) in (22), we can write

$$\boldsymbol{M}\boldsymbol{q}_\ell = \boldsymbol{q}_{\ell+1}\|\boldsymbol{r}_\ell\| + \sum_{i=1}^{\ell} h_{i,\ell}\boldsymbol{q}_i$$

$$= \sum_{i=1}^{\ell+1} h_{i,\ell}\boldsymbol{q}_i\,. \tag{24}$$

Hence, after $m$ iterations the Arnoldi iteration yields the relation

$$\boldsymbol{M}\boldsymbol{Q}_m = \boldsymbol{Q}_m\boldsymbol{H}_m + h_{m+1,m}\boldsymbol{q}_{m+1}\boldsymbol{e}_m^\mathsf{T}\,, \tag{25}$$

where $\boldsymbol{Q}_m = [\boldsymbol{q}_1,\ldots,\boldsymbol{q}_m]$ is the matrix of Arnoldi basis vectors, $\boldsymbol{H}_m \in \mathbb{R}^{m\times m}$ is an upper Hessenberg matrix with entries $h_{ij} = \boldsymbol{q}_i^\mathsf{T}\boldsymbol{M}\boldsymbol{q}_j$, and $\boldsymbol{e}_m$ is the $m$-th standard basis vector. The Arnoldi iteration is summarized in Algorithm 1.

Equation (25), known as the **Arnoldi relation**, shows that the eigenvalues of $\boldsymbol{H}_m$ approximate those of $\boldsymbol{M}$.

---

**Algorithm 1** The Arnoldi iteration for the computation of an orthonormal basis of a Krylov space

---

1: Let $\boldsymbol{M} \in \mathbb{R}^{n\times n}$. This algorithm computes an orthonormal basis for $\mathcal{K}_m(\boldsymbol{M}, \boldsymbol{x})$.
2: $\boldsymbol{q}_1 = \boldsymbol{x}/\|\boldsymbol{x}\|$;
3: **for** $\ell = 1,\ldots, m$ **do**
4:    $\boldsymbol{r} := \boldsymbol{M}\boldsymbol{q}_\ell$;
5:    **for** $i = 1,\ldots,\ell$ **do**
6:       $h_{i\ell} := \boldsymbol{q}_i^\mathsf{T}\boldsymbol{r}$   $\boldsymbol{r} := \boldsymbol{r} - h_{i\ell}\boldsymbol{q}_i$;
7:    **end for**
8:    $h_{\ell+1,\ell} := \|\boldsymbol{r}\|$;
9:    **if** $h_{\ell+1,\ell} = 0$ **then**
10:       **return** $(\boldsymbol{q}_1,\ldots,\boldsymbol{q}_\ell, \boldsymbol{H} \in \mathbb{R}^{\ell\times\ell})$
11:    **end if**
12:    $\boldsymbol{q}_{\ell+1} = \boldsymbol{r}/h_{\ell+1,\ell}$;
13: **end for**
14: **return** $(\boldsymbol{q}_1,\ldots,\boldsymbol{q}_{m+1}, \boldsymbol{H} \in \mathbb{R}^{m+1\times m})$

---

Specifically, assuming $h_{m+1,m}$ is small, using (25) $\boldsymbol{M}$ can be approximated as

$$\boldsymbol{M} \approx \boldsymbol{Q}_m\boldsymbol{H}_m\boldsymbol{Q}_m^\mathsf{T}\,. \tag{26}$$

Let $\boldsymbol{H}_m = \boldsymbol{V}\boldsymbol{\Sigma}\boldsymbol{V}^\mathsf{T}$ the eigendecomposition of $\boldsymbol{H}_m$. Then, the eigenvalues of $\boldsymbol{H}_m$ serve as an approximation of some of the eigenvalues of $\boldsymbol{M}$, and the corresponding eigenvectors of $\boldsymbol{M}$, denoted by $\boldsymbol{u}$, can be obtained as $\boldsymbol{u} = \boldsymbol{Q}_m\boldsymbol{v}$. Substituting this eigendecomposition into (26) yields the approximate eigendecomposition of $\boldsymbol{M}$:

$$\boldsymbol{M} \approx \boldsymbol{U}\boldsymbol{\Sigma}\boldsymbol{U}^\mathsf{T}, \tag{27}$$

where $\boldsymbol{U} = \boldsymbol{Q}_k\boldsymbol{V}$ and $\boldsymbol{U}^\mathsf{T}\boldsymbol{U} \approx \boldsymbol{I}$.

## B.2 Proposed Decentralized Arnoldi Iteration

For later use, we denote by $\boldsymbol{v}_\mathcal{I} = [v_i, \forall i \in \mathcal{I}]$ the entries of vector $\boldsymbol{v}$ indexed by the set $\mathcal{I}$.

We aim to use Arnoldi iteration to estimate the smallest $r$ eigenvalues of $\boldsymbol{L}_\mathcal{G}$ and their corresponding eigenvectors in a decentralized manner across clients while preserving privacy.

Each client knows only the incoming and outgoing connections to its local nodes[1] and does not have other knowledge about the subgraphs of other clients. Formally, Client $i$ knows the rows and columns

---

[1] The assumption that interconnections between clients are known, i.e., a node in a given client knows the existence of a node in another client and the edge connecting them, is both realistic and reflective of several

of the adjacency matrix $\boldsymbol{A}$ corresponding to its internal nodes $v \in \mathcal{V}_i$, i.e., $\boldsymbol{A}_{\mathcal{V}_i,:}$ and $\boldsymbol{A}_{:,\mathcal{V}_i}$, and subsequently the corresponding rows and columns of the degree matrix, $\boldsymbol{D}_{\mathcal{V}_i,:}$ and $\boldsymbol{D}_{:,\mathcal{V}_i}$.

In the Arnoldi iteration, clients need to collaboratively compute

$$\boldsymbol{q}_{\ell+1} = \frac{\boldsymbol{r}_\ell}{\|\boldsymbol{r}_\ell\|}, \quad \boldsymbol{r}_\ell = \boldsymbol{L}_\mathcal{G} \boldsymbol{q}_\ell - \sum_{i=1}^{\ell} h_{i,\ell} \boldsymbol{q}_i, \tag{28}$$

which follows from the Arnoldi update (22) with $\boldsymbol{M} = \boldsymbol{L}_\mathcal{G}$.

Carrying out (28) in a decentralized way requires each Client $i$ compute its local portion $\boldsymbol{r}_{\mathcal{V}_i}$ (for a generic vector $\boldsymbol{r}$). Effectively, this means performing the matrix-vector multiplication $\boldsymbol{b} = \boldsymbol{L}_\mathcal{G} \boldsymbol{q}$ (for a generic vector $\boldsymbol{q}$), where $\boldsymbol{b}, \boldsymbol{q} \in \mathbb{R}^n$ are $n$-dimensional vectors, and computing the coefficients $h_{i,\ell} = \boldsymbol{q}_i^\top \boldsymbol{L}_\mathcal{G} \boldsymbol{q}_\ell$ in a privacy-preserving way: Neither the clients nor the central server should be able to reconstruct the global vectors $\boldsymbol{b}$ or $\boldsymbol{q}$. To achieve this, we decompose $\boldsymbol{b}_{\mathcal{V}_i}$ as follows:

$$\begin{aligned}
\boldsymbol{b}_{\mathcal{V}_i} &= (\boldsymbol{L}_\mathcal{G} \boldsymbol{q})_{\mathcal{V}_i} \\
&= ((\boldsymbol{D} - \boldsymbol{A})\boldsymbol{q})_{\mathcal{V}_i} \\
&= (\boldsymbol{D}\boldsymbol{q} - \boldsymbol{A}\boldsymbol{q})_{\mathcal{V}_i} \\
&= (\boldsymbol{D}\boldsymbol{q})_{\mathcal{V}_i} - (\boldsymbol{A}\boldsymbol{q})_{\mathcal{V}_i} \\
&= \boldsymbol{D}_{\mathcal{V}_i \mathcal{V}_i} \boldsymbol{q}_{\mathcal{V}_i} - \boldsymbol{A}_{\mathcal{V}_i:} \boldsymbol{q} \\
&= \boldsymbol{D}_{\mathcal{V}_i, \mathcal{V}_i} \boldsymbol{q}_{\mathcal{V}_i} - \sum_{j=1}^{K} \boldsymbol{A}_{\mathcal{V}_i, \mathcal{V}_j} \boldsymbol{q}_{\mathcal{V}_j},
\end{aligned} \tag{29}$$

where $K$ is the number of clients.

We observe that the first term of (29), $\boldsymbol{D}_{\mathcal{V}_i, \mathcal{V}_i} \boldsymbol{q}_{\mathcal{V}_i}$, can be computed by Client $i$ using its local knowledge. However, the second term requires collaboration among clients, as $\boldsymbol{q}_{\mathcal{V}_j}$ for $j \neq i$ is unknown to Client $i$.

Since client $i$ only requires $\sum_{j=1}^{K} \boldsymbol{A}_{\mathcal{V}_i, \mathcal{V}_k} \boldsymbol{q}_{\mathcal{V}_j}$, clients can employ homomorphic encryption to securely compute this sum via the server. Specifically, each Client $j$ encrypts its local product $\boldsymbol{A}_{\mathcal{V}_i, \mathcal{V}_k} \boldsymbol{q}_{\mathcal{V}_j}$ and sends $\boldsymbol{h}_{\mathcal{V}_i}^{(j)} = \text{HE}(\boldsymbol{A}_{\mathcal{V}_i, \mathcal{V}_k} \boldsymbol{q}_{\mathcal{V}_j})$ to the central server, where $\text{HE}(\cdot)$ is a homomorphic encryption function. The server computes the encrypted sum $\boldsymbol{h}_{\mathcal{V}_i} = \sum_{j=1}^{K} \boldsymbol{h}_{\mathcal{V}_i}^{(j)}$ and sends $\boldsymbol{h}_{\mathcal{V}_i}$ to Client $i$. Finally, Client $i$ decrypts it to obtain the required sum $\sum_{j=1}^{K} \boldsymbol{A}_{\mathcal{V}_i, \mathcal{V}_j} \boldsymbol{q}_{\mathcal{V}_j}$ as $\sum_{j=1}^{K} \boldsymbol{A}_{\mathcal{V}_i, \mathcal{V}_k} \boldsymbol{q}_{\mathcal{V}_j} = \text{HD}(\boldsymbol{h}_{\mathcal{V}_i})$, where $\text{HD}(\cdot)$ is the homomorphic decryption function.

Through this approach, neither the central server nor any Client $i$ can reconstruct the components $\boldsymbol{A}_{\mathcal{V}_i, \mathcal{V}_k} \boldsymbol{q}_{\mathcal{V}_j}$ for $j \neq i$. Furthermore, as $\boldsymbol{b}_{\mathcal{V}_i} \in \mathbb{R}^{n_i}$ has dimension $n_i$ and the remaining $n - n_i$ entries of $\{\boldsymbol{q}_{\mathcal{V}_j} \mid \forall k \neq i\}$ are unknown to Client $i$, it cannot reconstruct $\{\boldsymbol{q}_{\mathcal{V}_j} \mid \forall j \neq i\}$ as long as $n - n_i \geq n_i$. The proposed decentralized Arnoldi iteration is detailed in Algorithm 2.

In Appendix C, we formally demonstrate that the proposed decentralized Arnoldi iteration prevents clients from inferring the internal subgraph structure of other clients, thereby ensuring FEDLAP preserves privacy.

### B.3 Computational Complexity of the Arnoldi Iteration

Computing the eigenvalues and eigenvectors of a graph Laplacian matrix is generally considered computationally expensive. However, FEDLAP circumvents this limitation by leveraging the Arnoldi iteration, a technique that is particularly efficient for sparse and low-rank graphs, which are common in real-world datasets.

real-world scenarios. This setting naturally arises in applications where edges originate locally but terminate in another client's subgraph, and the originating client must know the identifier of the destination node. For example, in banking, a bank records a transaction to a customer at another bank and therefore knows the recipient's identifier (e.g., IBAN). In anti-money laundering applications, this assumption is standard [14]. Also, in supply chains, a company places an order with a supplier managed by another organization and must identify the recipient entity. Moreover, this setting has been explicitly adopted in prior work on subgraph federated learning, including FedStruct and FedGCN, which further supports its practical relevance.

**Algorithm 2** The Decentralized Arnoldi algorithm for the computation of an orthonormal basis of a Krylov space

1: Let $\boldsymbol{A} \in \mathbb{R}^{n \times n}$. $K$ clients with client $i$ knowing $\boldsymbol{A}_{\mathcal{V}_i,:}$ and $\boldsymbol{A}_{:,\mathcal{V}_i}$ and $\boldsymbol{x}_{\mathcal{V}_i}$ for an input vector $\boldsymbol{x}$. This algorithm computes an orthonormal basis for $\mathcal{K}_m(\boldsymbol{L}_\mathcal{G}, \boldsymbol{x})$.

2: $\text{————————}\boldsymbol{Q}_{:,1} = \boldsymbol{x}/\|\boldsymbol{x}\|;\text{————————}$

3: **for** $i = 1, \ldots, K$ **do**

4:     Client $i$ sends $\text{HE}(\|\boldsymbol{x}_{\mathcal{V}_i}\|^2)$ to the server

5: **end for**

6: Server computes $\sum_{i=1}^{K} \text{HE}(\|\boldsymbol{x}_{\mathcal{V}_i}\|^2)$ and sends it to clients

7: Clients calculate $\|\boldsymbol{x}\| = \sqrt{\text{HD}(\sum_{i=1}^{K} \text{HE}(\|\boldsymbol{x}_{\mathcal{V}_i}\|^2))}$

8: $\boldsymbol{Q}_{\mathcal{V}_i,1} = \boldsymbol{x}_{\mathcal{V}_i}/\|\boldsymbol{x}\|;$

9: **for** iteration $\ell = 1, \ldots, m$ **do**

10:     $\text{————————————}\boldsymbol{r} := \boldsymbol{L}_\mathcal{G}\boldsymbol{Q}_{:,\ell};\text{————————}$

11:     **for** $i = 1, \ldots, K$ **do**

12:         **for** $k = 1, \ldots, K$ **do**

13:             Client $k$ sends $\boldsymbol{h}_{\mathcal{V}_i}^{(k)} = \text{HE}\left(\boldsymbol{A}_{\mathcal{V}_i,\mathcal{V}_k}\boldsymbol{Q}_{\mathcal{V}_k,\ell}\right)$ to the server

14:         **end for**

15:         Server does $\boldsymbol{h}_{\mathcal{V}_i} = \sum_{k=1}^{K} \boldsymbol{h}_{\mathcal{V}_i}^{(k)}$ and sends $\boldsymbol{h}_{\mathcal{V}_i}$ to client $i$

16:         Client $i$ calculates $\sum_{k=1}^{K} \boldsymbol{A}_{\mathcal{V}_i,\mathcal{V}_k}\boldsymbol{Q}_{\mathcal{V}_k,\ell} = \text{HD}(\boldsymbol{h}_{\mathcal{V}_i})$

17:         $\boldsymbol{r}_{\mathcal{V}_i} = \boldsymbol{D}_{\mathcal{V}_i\mathcal{V}_i}\boldsymbol{q}_{\mathcal{V}_i} - \sum_{k=1}^{K} \boldsymbol{A}_{\mathcal{V}_i,\mathcal{V}_k}\boldsymbol{Q}_{\mathcal{V}_k,\ell}$

18:     **end for**

19:     $\text{————————}h_{t\ell} := \boldsymbol{Q}_{:,t}^{\mathsf{T}}\boldsymbol{r}; \quad \boldsymbol{r} := \boldsymbol{r} - h_{t\ell}\boldsymbol{Q}_{:,t}\text{————————}$

20:     **for** $t = 1, \ldots, \ell$ **do**

21:         **for** $i = 1, \ldots, K$ **do**

22:             Client $i$ sends $\text{HE}(\boldsymbol{Q}_{\mathcal{V}_i,t}^{\mathsf{T}}\boldsymbol{r}_{\mathcal{V}_i})$ to the server

23:         **end for**

24:         Server computes $\sum_{i=1}^{K} \text{HE}(\boldsymbol{Q}_{\mathcal{V}_i,t}^{\mathsf{T}}\boldsymbol{r}_{\mathcal{V}_i})$ and sends it to clients

25:         Clients calculate $h_{t\ell} = \text{HD}(\sum_{i=1}^{K} \text{HE}(\boldsymbol{Q}_{\mathcal{V}_i t}^{\mathsf{T}}\boldsymbol{r}_{\mathcal{V}_i}))$

26:         $\boldsymbol{r}_{\mathcal{V}_i} := \boldsymbol{r}_{\mathcal{V}_i} - h_{t\ell}\boldsymbol{Q}_{\mathcal{V}_i,t};$

27:     **end for**

28:     $\text{————————————}h_{\ell+1,\ell} := \|\boldsymbol{r}\|;\text{————————}$

29:     **for** $i = 1, \ldots, K$ **do**

30:         Client $i$ sends $\text{HE}(\|\boldsymbol{r}_{\mathcal{V}_i}\|^2)$ to the server

31:     **end for**

32:     Server computes $\sum_{i=1}^{K} \text{HE}(\|\boldsymbol{r}_{\mathcal{V}_i}\|^2)$ and sends it to clients

33:     Clients do $\|\boldsymbol{r}\| = \sqrt{\text{HD}(\sum_{i=1}^{K} \text{HE}(\|\boldsymbol{r}_{\mathcal{V}_i}\|^2))}$

34:     $h_{\ell+1,\ell} := \|\boldsymbol{r}\|;$

35:     **if** $h_{(\ell+1)\ell} = 0$ **then**

36:         %Found invariant subspace%

37:         **for** $i = 1, \ldots, K$ **do**

38:             **return** $(\boldsymbol{Q}_{\mathcal{V}_i,:} \in \mathbb{R}^{n_i \times \ell}, \boldsymbol{H} \in \mathbb{R}^{\ell \times \ell})$

39:         **end for**

40:     **end if**

41:     $\text{————}\boldsymbol{Q}_{:,\ell+1} = \boldsymbol{r}/h_{(\ell+1)\ell};\text{————————}$

42:     **for** $i = 1, \ldots, K$ **do**

43:         $\boldsymbol{Q}_{\mathcal{V}_i,(\ell+1)} = \boldsymbol{r}_{\mathcal{V}_i}/h_{\ell+1,\ell};$

44:     **end for**

45: **end for**

46: **for** $i = 1, \ldots, K$ **do**

47:     **return** $(\boldsymbol{Q}_{\mathcal{V}_i,:} \in \mathbb{R}^{n_i \times m}, \boldsymbol{Q}_{\mathcal{V}_i,(m+1)} \in \mathbb{R}^{n_i}, \boldsymbol{H} \in \mathbb{R}^{m \times m}, h_{(m+1)m})$

48: **end for**

The computational complexity of the Arnoldi iteration for extracting the top $r$ eigenvectors of a sparse matrix of size $n \times n$ is primarily determined by two operations:

1. **Matrix-vector multiplication:** Each iteration involves multiplying the sparse Laplacian matrix with a vector. This operation has a cost of $O(n \cdot \bar{d})$, where $\bar{d}$ is the average degree of the graph.

2. **Orthogonalization:** The newly computed vector must be orthogonalized against all previous vectors, requiring $O(n \cdot r^2)$ operations over $r$ iterations.

Thus, the total computational complexity after $r$ Arnoldi iterations is:

$$O(r \cdot n \cdot \bar{d} + n \cdot r^2)$$

In practical scenarios with sparse graphs (i.e., $\bar{d} \ll r$), the orthogonalization step dominates, resulting in an effective complexity of $O(n \cdot r^2)$.

We illustrate this with two widely used benchmark datasets:

- **ogbn-arxiv** ($n = 169{,}343$, $\bar{d} = 13.7$):
  - For $r = 100$: $O(169{,}343 \times 100^2) \approx 1.69 \times 10^9$ operations
  - For $r = 200$: $O(169{,}343 \times 200^2) \approx 6.77 \times 10^9$ operations
- **ogbn-products** ($n = 2{,}449{,}029$, $\bar{d} = 50.5$):
  - For $r = 100$: $O(2{,}449{,}029 \times 100^2) \approx 2.45 \times 10^{10}$ operations
  - For $r = 200$: $O(2{,}449{,}029 \times 200^2) \approx 9.80 \times 10^{10}$ operations

These computations are feasible with standard hardware and can be further optimized using distributed implementations. Overall, the Arnoldi method offers a scalable and communication-efficient strategy for spectral approximation in federated graph settings.

### B.4  Learning in FEDLAP+ with the Arnoldi Iteration

After $r$ iterations of the decentralized Arnoldi iteration introduced in Appendix B.2, each Client $i$ obtains matrices $\boldsymbol{Q}_{\mathcal{V}_i,:} \in \mathbb{R}^{n_i \times r}$ and $\boldsymbol{H}_r \in \mathbb{R}^{r \times r}$ (see also Algorithm 2). Since $\boldsymbol{H}_r$ is shared among all clients, each client can decompose it as

$$\boldsymbol{H}_r = \boldsymbol{V}\boldsymbol{\Sigma}\boldsymbol{V}^\top \,, \tag{30}$$

where $\boldsymbol{\Sigma} \in \mathbb{R}^{r \times r}$ is the diagonal matrix of eigenvalues of $\boldsymbol{H}_r$ and $\boldsymbol{V} \in \mathbb{R}^{r \times r}$ the matrix of corresponding eigenvectors.

Each client $i$ can then compute

$$\boldsymbol{U}_{\mathcal{V}_i} = \boldsymbol{Q}_{\mathcal{V}_i,:}\boldsymbol{V} \,. \tag{31}$$

With this, an approximate eigendecomposition of the graph Laplacian can be written as (see (27))

$$\boldsymbol{L}_{\mathcal{G}} \approx \boldsymbol{U}\boldsymbol{\Sigma}\boldsymbol{U}^\top \,, \tag{32}$$

where $\boldsymbol{U}$ is formed by concatenating the matrices $\boldsymbol{U}_{\mathcal{V}_i}$.

FEDLAP+ uses this approximation of the Laplacian for learning. Specifically, for node $v$ in Client $i$, when using the decentralized Arnoldi iteration to approximate the graph Laplacian, FEDLAP+ performs node prediction as

$$\hat{\boldsymbol{y}}_v = \operatorname{softmax}\!\left( f_{\boldsymbol{\theta}_\mathrm{f}}(\boldsymbol{X}_i, \mathcal{E}_i, v) + g_{\boldsymbol{\theta}_\mathrm{s}}(\boldsymbol{U}_{v,:}\boldsymbol{W}) \right),$$

$$\mathcal{L}(\boldsymbol{\theta}, \boldsymbol{W}) = \mathcal{L}_\mathrm{c}(\boldsymbol{\theta}) + \lambda_\mathrm{reg}\frac{\operatorname{Tr}(\boldsymbol{W}^\top\boldsymbol{\Sigma}\boldsymbol{W})}{\operatorname{Tr}(\boldsymbol{W}^\top\boldsymbol{W})} \tag{33}$$

The model parameters $\boldsymbol{W}$ are updated as

$$\boldsymbol{W} \leftarrow \boldsymbol{W} - \lambda_\mathrm{w}\nabla_{\boldsymbol{W}}\mathcal{L}(\boldsymbol{\theta}, \boldsymbol{W}) \,. \tag{34}$$

# C   Privacy Analysis of FEDLAP+

In this appendix, we provide detailed derivations supporting the privacy analysis presented in Section 5 of the main paper.

Our focus is on the **offline phase** of FEDLAP+, where clients collaboratively estimate the eigenvectors of the graph Laplacian using the decentralized Arnoldi iteration (see Appendix B.2 and Algorithm 2). Unlike the online phase—which involves standard model updates and can be protected using established privacy-enhancing techniques such as differential privacy or secure aggregation—the offline phase involves sharing linear-algebraic components derived from local graph structure. This creates novel privacy challenges that warrant careful analysis.

In particular, we aim to quantify the ability of an attacker to infer *local connections* within another client. To this end, we consider a **worst-case scenario** in which the system consists of only two clients: Client 1 is the target and Client 2 is the attacker. The attacker attempts to infer whether there is an edge between two nodes $u, v \in \mathcal{V}_1$, the node set of Client 1. This is formulated as a binary hypothesis test:

- $H_0$: no edge exists between $u$ and $v$, i.e., $A_{uv} = 0$,
- $H_1$: an edge exists between $u$ and $v$, i.e., $A_{uv} = 1$.

We study the distribution of the *log-likelihood ratio (LLR)* associated with this test and analyze how well the attacker can distinguish between the two hypotheses.

**Structure of this appendix.**    The remainder of this section is organized as follows:

- Section C.1 introduces the attack model.
- Section C.2 introduces the assumptions made for the analysis.
- Section C.3 provides the full proof of Theorem 1, which characterizes the distribution of the LLRs under both hypotheses.
- Section C.4 contains the proof of Corollary 1, which provides an expression for the Kullback–Leibler divergence between the two LLR distributions and analyzes its dependence on key parameters such as the truncation rank $r$, the number of nodes $n$, and the connection probability $p$.
- Section C.5 derives the attacker's true positive rate (TPR) and false positive rate (FPR), and uses these to compute the corresponding precision and recall. This analysis enables us to quantify the privacy guarantees offered by FEDLAP+.

## C.1   Attacker model

This appendix gives a detailed account of what the attacker can observe in the decentralized Arnoldi protocol.

**What the attacker observes.**    Recall the local block identity (Equation (16)):

$$\boldsymbol{b}_{\mathcal{V}_i} \;=\; \boldsymbol{D}_{\mathcal{V}_i, \mathcal{V}_i}\, \boldsymbol{q}_{\mathcal{V}_i} \;-\; \sum_{j=1}^{K} \boldsymbol{A}_{\mathcal{V}_i, \mathcal{V}_j}\, \boldsymbol{q}_{\mathcal{V}_j}. \tag{35}$$

From the secure aggregation step, the attacker (client $i$) receives only the *aggregated* vector

$$\boldsymbol{\tau}_{\mathcal{V}_i} \;=\; \sum_{j=1}^{K} \boldsymbol{A}_{\mathcal{V}_i, \mathcal{V}_j}\, \boldsymbol{q}_{\mathcal{V}_j}\,.$$

The attacker also knows the adjacency blocks $\boldsymbol{A}_{\mathcal{V}_i, \mathcal{V}_j}$ that correspond to its own outgoing/incoming inter-client edges. Thus the attacker has $n_i$ linear constraints:

$$\boldsymbol{A}_{i, \neg i}\, \boldsymbol{q}_{\neq i} = \boldsymbol{\tau}_{\mathcal{V}_i}, \qquad \boldsymbol{A}_{i, \neg i} \triangleq \big[\boldsymbol{A}_{\mathcal{V}_i, \mathcal{V}_1} \;\cdots\; \boldsymbol{A}_{\mathcal{V}_i, \mathcal{V}_{i-1}} \;\boldsymbol{A}_{\mathcal{V}_i, \mathcal{V}_{i+1}} \;\cdots\; \boldsymbol{A}_{\mathcal{V}_i, \mathcal{V}_K}\big],$$

where $\boldsymbol{q}_{\neq i} = [\boldsymbol{q}_{\mathcal{V}_j}]_{j \neq i}$ is the stacked vector of unknown spectral blocks of other clients. Since $n_i < n - n_i$ in typical settings, the system is underdetermined and infinitely many $\boldsymbol{q}_{\neg i}$ satisfy the

observed equations. Consequently, the attacker can only produce an estimate $\breve{q}_{\neq i}$. Collecting the estimates of $\boldsymbol{Q}_{\neg i,:} = [\boldsymbol{Q}_{\mathcal{V}_j}]_{j \neq i}$ as $\breve{\boldsymbol{Q}} \in \mathbb{R}^{n-n_i \times r}$, we can write

$$\|\breve{\boldsymbol{Q}} - \boldsymbol{Q}_{\neg i,:}\| \leq \sigma. \tag{36}$$

Using the Arnoldi relation (14) and the public matrix $\boldsymbol{H}_r$, an attacker with estimate $\hat{\boldsymbol{Q}}_r$ forms

$$\boldsymbol{U} \triangleq \boldsymbol{D}_{\neg i}\breve{\boldsymbol{Q}} + \boldsymbol{A}_{\neg i, \mathcal{V}_i}\boldsymbol{Q}_{\mathcal{V}_i,:} - \breve{\boldsymbol{Q}}\boldsymbol{H}_r.$$

Hence, the attacker faces the reconstruction problem

$$\boldsymbol{U} \approx \breve{\boldsymbol{A}}\breve{\boldsymbol{Q}}, \tag{37}$$

where $\breve{\boldsymbol{A}} = \boldsymbol{A}_{\neg i, \neg i}$ is the unknown target adjacency block and $\breve{\boldsymbol{Q}} = \hat{\boldsymbol{Q}}_{\neg i,:}$ is noisy (the attacker's estimate). Note that equality in (37) holds only when $\sigma = 0$ and $h_{r+1,r} = 0$. The attacker must also know $\boldsymbol{D}_{\neg i, \neg i}$ to calculate $\boldsymbol{U}$. The attacker then performs the following steps: (i) obtain $\boldsymbol{U} = \breve{\boldsymbol{Q}}_r\boldsymbol{H}_r$, (ii) evaluate the log-likelihood ratio $\mathrm{LLR}_{u,v}$ for the two hypotheses using $\boldsymbol{U}$, and (iii) decide $H_1$ whenever $\mathrm{LLR}_{u,v} \geq \gamma$ for some threshold $\gamma \in \mathbb{R}$.

## C.2 Assumptions

### C.2.1 Modeling assumptions

To enable a tractable and rigorous analysis, we assume that the graph connections follow a Bernoulli distribution. This setup corresponds to a simplified instance of the Stochastic Block Model (SBM), a common generative model for graphs with community structure. In the SBM, the probability of an edge between two nodes depends on whether they belong to the same community ($p$) or different communities ($q$). Specifically, $p$ is the probability of an intra-community edge and $q$ is the probability of an inter-community edge. In our analysis we consider the case where the attacker assumes $p = q$, meaning all node pairs are connected independently with equal probability. While this assumption may not perfectly reflect community-structured real-world graphs, it provides a conservative and attacker-agnostic baseline. In realistic scenarios, adversaries are unlikely to know the exact community assignments, making the $p = q$ setting a reasonable approximation for worst-case analysis. Moreover, both the SBM and Bernoulli model are widely adopted in the graph learning literature as analytical tools, allowing us to derive privacy guarantees that remain meaningful under minimal structural assumptions.

### C.2.2 Worst-case scenario with two clients

We assume a scenario with two clients, where Client 1 is the target and Client 2 is a potentially malicious client attempting to infer private connections within Client 1. This models the worst-case setting where all other clients collude against a single target client.

### C.2.3 Low-rank approximation of adjacency matrix

The attacker observes a low-rank approximation

$$\boldsymbol{U} \approx \breve{\boldsymbol{A}}\breve{\boldsymbol{Q}}, \tag{38}$$

To simplify the analysis, in favor of the attacker we assume the equation holds with equality and therefore $\sigma = 0$ and $h_{r+1,r} = 0$. However, the attacker cannot reconstruct the exact adjacency matrix $\breve{\boldsymbol{A}}$ from this observation, even with full knowledge of $\breve{\boldsymbol{Q}}$.

Note that realistic adjacency matrices include clusters and are typically well-approximated by a low rank matrix [31]. Hence, even with full knowledge of $\breve{\boldsymbol{Q}}$, $\breve{\boldsymbol{A}}$ cannot be uniquely determined by observing $\boldsymbol{U}$.

### C.2.4 Delocalization and Orthogonality of Eigenvectors

To derive the analytical form of the privacy guarantees in Corollary 1, we assume that the columns of $\breve{\boldsymbol{Q}}$ are approximately orthogonal, i.e., $\breve{\boldsymbol{Q}}^\top \breve{\boldsymbol{Q}} \approx \boldsymbol{I}_r$, and that $\breve{\boldsymbol{Q}}$ is *delocalized*, meaning its columns

are spread uniformly over the unit sphere. This implies $|\check{\boldsymbol{Q}}_{v,:}|^2 \approx r/n$ for all $v \in \mathcal{V}_1$, where $r$ is the truncation rank and $n$ is the number of nodes in Client1.

These assumptions are grounded in empirical observations of spectral properties in real-world graphs, particularly under stochastic models such as the SBM and random regular graphs. However, we stress that they are not necessary for our privacy guarantees to hold. They are used purely to simplify the derivations and enable closed-form analysis.

Assuming delocalization and orthogonality gives the attacker more power than in most realistic settings. For instance, since the actual number of nodes is $n = n_1 + n_2 > n_1$, the true norm $\|\check{\boldsymbol{Q}}_{v,:}\|^2$ is often smaller than $r/n$, which decreases the attacker's ability to distinguish between hypotheses. As suggested by the KL divergence expression in Corollary 1 (see also (55) below), a smaller $r/n$ reduces statistical distinguishability, thereby enhancing privacy. Thus, our assumptions result in a conservative (i.e., worst-case) privacy analysis, further highlighting the robustness of our guarantees.

### C.2.5 Central Limit Theorem applicability

Lemma C.1 shows that the multivariate Lindeberg Central Limit Theorem (CLT) holds for our setting.

To address finite-sample effects, we refine this analysis using the multivariate Berry–Esseen theorem [32]. By Lemma C.2, the deviation of the empirical LLR distribution from the Gaussian limit scales as $\mathrm{Error}_{\mathrm{CLT}} = O(1/\sqrt{np})$, ensuring the validity of the CLT approximation even for moderate-sized graphs.

This bound clearly shows that the CLT approximation improves rapidly with larger $n$ or denser graphs (larger $np$). Even for moderate-size real-world graphs, where $p$ is small but $n$ is in the thousands, the approximation remains accurate.

Importantly, this assumption of large $n$ is used only to simplify the derivation of the LLR distribution; it does not weaken privacy guarantees for smaller graphs. In practice, the attacker's real-world inference capability is weaker than predicted by the asymptotic bound. As confirmed in our experiments (see Fig. 4), the theoretical bound remains conservative, and FEDLAP+ continues to provide strong privacy even for finite, moderately sized graphs.

**Lemma C.1.** *For $i \in [1, n]$, let $\boldsymbol{c}_i \in \mathbb{R}^r$ where $\|\boldsymbol{c}_i\|^2 = \mathcal{O}(1/n)$, and let $B_i \sim Ber(p)$, $p \in [0, 1]$. Define the random vector $\boldsymbol{y} = \sum_{i=1}^{n} B_i \boldsymbol{c}_i$. Then, for large $n$, we have*

$$\boldsymbol{y} \sim \mathcal{N}(p\mathbf{1}^\top \boldsymbol{C}, p(1-p)\boldsymbol{C}^\top \boldsymbol{C}) \tag{39}$$

*where $\boldsymbol{C} = [\boldsymbol{c}_1, \ldots, \boldsymbol{c}_n]^\top \in \mathbb{R}^{n \times r}$*

*Proof.* Let $\boldsymbol{\mu}_i = \mathbb{E}[B_i \boldsymbol{c}_i] = p\boldsymbol{c}_i$, and define the centered random variable $\tilde{\boldsymbol{y}}_i := (B_i - p)\boldsymbol{c}_i$. To invoke the multivariate Lindeberg CLT, we verify the Lindeberg condition:

$$\frac{1}{n} \sum_{i=1}^{n} \mathbb{E}\left[\|\tilde{\boldsymbol{y}}_i\|^2 \mathbb{1}(\|\tilde{\boldsymbol{y}}_i\| \geq \epsilon\sqrt{n})\right] \to 0 \text{ as } n \to \infty. \tag{40}$$

Since $B_i - p \in \{-p, 1-p\}$, we have $\|\tilde{\boldsymbol{y}}_i\| \leq \max(p, 1-p)\|\boldsymbol{c}_i\| = \mathcal{O}(1/\sqrt{n})$. Hence, (40) is upper-bounded as

$$\frac{1}{n} \sum_{i=1}^{n} \mathbb{E}\left[\|\tilde{\boldsymbol{y}}_i\|^2 \mathbb{1}(\|\tilde{\boldsymbol{y}}_i\| \geq \epsilon\sqrt{n})\right] \leq \frac{1}{n} \sum_{i=1}^{n} \|\boldsymbol{c}_i\|^2 \mathbb{E}\left[\mathbb{1}(\|\tilde{\boldsymbol{y}}_i\| \geq \epsilon\sqrt{n})\right] = \mathcal{O}(1/n) \to 0 \text{ as } n \to \infty. \tag{41}$$

Thus, the Lindeberg condition is satisfied. Since the total covariance is

$$\sum_{i=1}^{n} \mathrm{Cov}(\tilde{\boldsymbol{y}}_i) = p(1-p) \sum_{i=1}^{n} \boldsymbol{c}_i \boldsymbol{c}_i^\top = p(1-p)\boldsymbol{C}^\top \boldsymbol{C}, \tag{42}$$

we conclude the proof by invoking the multivariate Lindeberg CLT.

$\square$

**Lemma C.2** (Berry–Esseen bound for Bernoulli graph models). *Let $\{A_{uj}\}_{j=1}^{n}$ be independent Bernoulli$(p)$ random variables and define the normalized zero-mean vector*

$$x = \frac{1}{\sqrt{np(1-p)}}(A_{u,:} - p\mathbf{1})^{\top}Q,$$

*where $Q \in \mathbb{R}^{n \times r}$ is an orthonormal matrix satisfying $Q^{\top}Q = I_r$. Then $\mathbb{E}[x] = 0$ and $\mathrm{Cov}(x) = I_r$.*

*Let $\Phi_r$ denote the cumulative distribution function (CDF) of the $r$-dimensional standard normal distribution. Then, by the multivariate Berry–Esseen theorem [32], the deviation of the distribution of $x$ from the Gaussian limit satisfies*

$$\sup_{x \in \mathbb{R}^r} \left| \mathbb{P}[x \leq x] - \Phi_r(x) \right| \leq C \frac{\mathbb{E}[|A_{uj} - p|^3]}{(np(1-p))^{3/2}} = O\left(\frac{1}{\sqrt{np(1-p)}}\right),$$

*where $C > 0$ is an absolute constant independent of $n, p, r$. In the sparse-graph regime with small $p$, this simplifies to*

$$\mathrm{Error}_{\mathrm{CLT}} = O\left(\frac{1}{\sqrt{np}}\right). \tag{43}$$

*Proof.* Each coordinate of $x$ is a normalized sum of i.i.d. centered Bernoulli$(p)$ variables with variance $p(1-p)$. The univariate Berry–Esseen bound implies convergence to normality at rate $O(1/\sqrt{np(1-p)})$. Since $Q$ is orthonormal, linear combinations of these coordinates preserve the same rate in the multivariate case [32]. For sparse graphs ($p \ll 1$), the factor $(1-p)$ is absorbed into the constant, yielding (43). □

### C.3 Proof of Theorem 1

Following the assumptions in Appendix C.2.1, let the connections in $\breve{A} \in \{0,1\}^{n_1 \times n_1}$ be drawn independently from a Bernoulli distribution with parameter $p$. Based on the attack model in (38), the attacker's goal is to estimate specific entries $\breve{A}_{uv}$ to infer connections between nodes $u$ and $v$ within Client 1. Using Bayes, we write the posterior distribution of $\breve{A}_{uv}$ as

$$P(\breve{A}_{uv} = 1|U) = \frac{pP(U|\breve{A}_{uv} = 1)}{pP(U|\breve{A}_{uv} = 1) + (1-p)P(U|\breve{A}_{uv} = 0)}. \tag{44}$$

From (38), we note that

$$U_{u,:} = \sum_i \breve{A}_{ui}\breve{Q}_{i,:}. \tag{45}$$

Hence, each row $U$ is given by a sum of scaled independent Bernoulli random variables and $\|\breve{Q}_{i,:}\|^2 = \mathcal{O}(1/n)$. Therefore, Lemma C.1 applies and we can approximate the distribution $U_{u,:}$ as

$$U_{u,:} \sim \mathcal{N}(\mu, \Sigma), \tag{46}$$

where $\mu = p\mathbf{1}^{\top}\breve{Q}$ and $\Sigma = p(1-p)\breve{Q}^{\top}\breve{Q}$. By using (46) and by noting that $\breve{A}_{uv}$ only influences row $u$ in $U$, we find that

$$U_{u,:}|\breve{A}_{uv} = 1 \sim \mathcal{N}(\mu, \Sigma) \tag{47}$$

$$U_{u,:}|\breve{A}_{uv} = 0 \sim \mathcal{N}(\mu - \breve{Q}_{v,:}, \Sigma) \tag{48}$$

which, after some algebraic manipulations, results in the LLR

$$\mathrm{LLR}(\breve{A}_{uv}) = \log\left(\frac{P(U|\breve{A}_{uv} = 1)}{P(U|\breve{A}_{uv} = 0)}\right) \tag{49}$$

$$= (U_{u,:} - \mu + \frac{1}{2}\breve{Q}_{v,:})\Sigma^{-1}\breve{Q}_{v,:}^{\top}. \tag{50}$$

By using (47)–(48) and noting that (50) is a linear transformation of a Gaussian vector under the two hypotheses, we obtain

$$\text{LLR}(\breve{A}_{uv})|\breve{A}_{uv} = 1 \sim \mathcal{N}\left(\frac{1}{2}\alpha, \alpha\right) \tag{51}$$

$$\text{LLR}(\breve{A}_{uv})|\breve{A}_{uv} = 0 \sim \mathcal{N}\left(-\frac{1}{2}\alpha, \alpha\right), \tag{52}$$

where $\alpha = \breve{Q}_{v,:}\Sigma^{-1}\breve{Q}_{v,:}^{\mathsf{T}}$. This concludes the proof.

## C.4 Proof of Corollary 1

Based on the orthogonality assumption in C.2.4, the columns of $\breve{Q}$ are orthogonal. Therefore,

$$\Sigma^{-1} \approx \frac{1}{p(1-p)}I_r. \tag{53}$$

Also, based on the delocalized assumption in C.2.4, $\breve{Q}$ has delocalized rows, and it follows $\|\breve{Q}_{:,v}\|^2 \approx r/n$. Therefore, we can approximate $\alpha$ in Theorem 1 as

$$\alpha \approx \frac{1}{p(1-p)}\|\breve{Q}_{v,:}\|^2 = \frac{r}{np(1-p)}. \tag{54}$$

Note that the approximation of $\alpha$ is independent of $u$ and $v$.

Next, we consider the KL divergence between the two LLR distributions. Noting that the LLR distributions in Theorem 1 follow Normal distributions with the same variance, we have that

$$D_{\text{KL}}\left(\Pr\left(\text{LLR}(\breve{A}_{uv}) \mid \breve{A}_{uv} = 1\right) \,\|\, \Pr\left(\text{LLR}(\breve{A}_{uv}) \mid \breve{A}_{uv} = 0\right)\right) = \frac{\alpha}{2} \approx \frac{r}{2np(1-p)}, \tag{55}$$

where the last step follows from (54). This concludes the proof.

## C.5 Attack Performance and Privacy Guarantees

In this appendix, we derive the TPR and FPR for the attacker and discuss the resulting privacy guarantees.

We consider the LLR distributions for a given node pair $(u, v)$ under the two hypotheses. From Theorem 1, we have

$$H1 : LLR_{u,v} \sim \mathcal{N}\left(\frac{\alpha}{2}, \alpha\right) \tag{56}$$

$$H0 : LLR_{u,v} \sim \mathcal{N}\left(-\frac{\alpha}{2}, \alpha\right). \tag{57}$$

Using this, for a given threshold $\gamma \in \mathbb{R}$, we can derive the true positive rate (TPR) and false positive rate (FPR) as

$$\text{TPR} = P(\text{LLR}_{u,v} > \gamma \mid H_1)) = 1 - \Phi\left(\frac{\gamma - \frac{\alpha}{2}}{\sqrt{\alpha}}\right) \tag{58}$$

$$\text{FPR} = P(\text{LLR}_{u,v} > \gamma \mid H_0)) = 1 - \Phi\left(\frac{\gamma + \frac{\alpha}{2}}{\sqrt{\alpha}}\right), \tag{59}$$

where $\Phi(x)$ is the cumulative distribution function of the standard normal Gaussian distribution.

Real world graphs are typically sparse. Hence, there will be a strong imbalance between the two hypotheses. For this reason, we assess the attacker performance via precision and recall. The precision (P) and recall (R) can be expressed as

$$\text{P} = \frac{p\text{TPR}}{p\text{TPR} + (1-p)\text{FPR}}, \tag{60}$$

$$\text{R} = \text{TPR}. \tag{61}$$

Together, precision and recall measure the attacker's ability to correctly infer which pairs of nodes are connected. Given the distributions of TPR and FPR under our worst-case attacker model, one can compute these values and generate the corresponding **precision–recall curves**. In Fig. 3 in the main paper, we show this relationship for varying values of the truncation rank $r$, number of nodes $n$, and connection probability $p$.

Importantly, for any fixed $n$ and $p$, our analysis shows that it is possible to select a value of $r$ such that

$$P + R \leq 1 \,.$$

This inequality is a key indicator of privacy in our setting. Intuitively, when the sum of precision and recall falls below one, the attacker performs *worse than trivial guessing*. For example:

- If the attacker guesses all node pairs are connected, they achieve Recall $= 1$ and Precision $\approx 0$.
- If the attacker guesses all node pairs are disconnected, they achieve Precision $= 1$ and Recall $\approx 0$.

In both cases, $P + R \approx 1$. Thus, if $P + R \leq 1$, the attacker's best strategy reduces to guessing either everything is connected or nothing is—neither of which reveals any meaningful information about individual inter-client connections. This result underscores the strong privacy guarantees of FEDLAP+ under the analyzed threat model.

## C.6 Privacy analysis of Subgraph Federated Learning methods

In this section, we provide a more detailed discussion of the privacy guarantees offered by FEDLAP and contrast them with those of existing SFL approaches, notably FEDSTRUCT and FEDGCN. We also discuss the challenges in conducting a formal privacy analysis for these baselines.

### C.6.1 Two-Phase Privacy Perspective

To structure our privacy analysis, we divide FEDLAP into two conceptual phases:

- **Offline phase**: This phase occurs once before training begins and is responsible for computing structural components using the Arnoldi iteration. It involves exchanging partial results of matrix-vector multiplications (i.e., $\boldsymbol{Aq}$) but does not share raw adjacency or feature information.
- **Online phase**: This corresponds to standard FL training and introduces no additional privacy risks beyond those already known in FL. Any conventional privacy-preserving mechanism commonly used in FL—such as differential privacy or secure aggregation—can be directly applied in this phase.

As a result, the main privacy concern is restricted to the offline phase, and in our paper, we provide a formal analysis of this phase under a worst-case scenario. Even assuming a strong attacker with access to all intermediate values (e.g., $\boldsymbol{U} = \boldsymbol{\breve{A}\breve{Q}}$ and $\boldsymbol{Q}$), we have demonstrated in Appendix C that inferring intra-client edges becomes infeasible under reasonable sparsity and rank conditions. This analysis establishes FEDLAP's privacy guarantees on a firm theoretical foundation.

### C.6.2 Comparison with FEDGCN and FEDSTRUCT

No formal privacy analysis exists for FEDSTRUCT or FEDGCN, making FEDLAP especially appealing. Furthermore, applying our privacy framework to these methods is not straightforward due to the nature of the information they exchange:

- **FEDGCN** shares aggregated node features—typically the sum of features of neighboring nodes. As shown in [16], even secure aggregation offers weak protection against membership inference attacks. Moreover, when node features are sparse and structured (e.g., binary encodings of names), reconstruction becomes alarmingly feasible.

  Consider a toy example where node features encode ASCII binary representations of account names:

- Alice: [01000001, 01101100, 01101001, 01100011, 01100101]
- Bob: [01000010, 01101111, 01100010, 00000000, 00000000]
- Sum: [000000011, 000000011, 000001011, 001100011, 001100101]

An attacker with access to this aggregated sum can precompute the sum of known character encodings and match the result, effectively inferring sensitive identities. When nodes participate in multiple aggregations, the adversary obtains overlapping constraints, compounding the privacy risk.

- **FEDSTRUCT** introduces a large learnable structure matrix $S$, which is iteratively updated and shared across clients during training. This makes the privacy analysis highly nontrivial. Although its offline setup phase may potentially be analyzed using our black-box approach, the online phase presents serious challenges. The continuous sharing of gradients with respect to $S$, and the exposure of global model updates, pose significant risks that are difficult to quantify formally. The authors of FEDSTRUCT acknowledge this by including an attack in their Appendix G.1, which demonstrates concrete leakage scenarios.

Despite these challenges, we provide the following intuitive arguments for why FEDLAP offers stronger privacy guarantees:

- FEDLAP reduces the need for direct structural or feature sharing, instead relying on local matrix-vector computations through Arnoldi iteration.
- The structural information shared is limited and one-time (offline), unlike FEDSTRUCT, which exposes evolving parameters over training.
- The decomposition used in FEDLAP+ allows for distributing only local structural components (i.e., relevant rows of $U$), further minimizing exposure.

## D  Convergence guarantee of FEDLAP+

We analyze the smoothness of the spectral regularizer to establish the convergence guarantee of FEDLAP+ under the standard FEDAVG framework. Our online loss is defined as

$$L(\boldsymbol{\theta}) = L_c(\boldsymbol{\theta}) + \lambda_{\text{reg}} R(\boldsymbol{W}), \qquad R(\boldsymbol{W}) = \frac{\text{Tr}(\boldsymbol{W}^\top \boldsymbol{\Lambda} \boldsymbol{W})}{\text{Tr}(\boldsymbol{W}^\top \boldsymbol{W})}, \tag{62}$$

where $L_c(\boldsymbol{\theta})$ is the supervised loss (e.g., cross-entropy), $\boldsymbol{\Lambda}$ is the diagonal matrix of Laplacian eigenvalues, and $\boldsymbol{W}$ contains the spectral coefficients. To ensure the convergence of FEDAVG, we examine the smoothness of the regularizer $R(\boldsymbol{W})$.

Since $R(\boldsymbol{W})$ is scale-invariant ($R(\alpha \boldsymbol{W}) = R(\boldsymbol{W})$ for any $\alpha > 0$), we normalize $\boldsymbol{W}$ to have unit Frobenius norm ($\|\boldsymbol{W}\|_F = 1$) after each local update. On the unit sphere, the gradient of $R(\boldsymbol{W})$ is given by

$$\nabla_{\boldsymbol{W}} R(\boldsymbol{W}) = 2(\boldsymbol{\Lambda} \boldsymbol{W} - \boldsymbol{W} \, \text{Tr}(\boldsymbol{W}^\top \boldsymbol{\Lambda} \boldsymbol{W})). \tag{63}$$

This gradient is Lipschitz-continuous. For any $\boldsymbol{W}_1, \boldsymbol{W}_2$ with $\|\boldsymbol{W}_1\|_F = \|\boldsymbol{W}_2\|_F = 1$, and using $\|\boldsymbol{\Lambda}\|_2 = \lambda_{\max}$, we have

$$\|\nabla R(\boldsymbol{W}_1) - \nabla R(\boldsymbol{W}_2)\| \le 8\lambda_{\max} \|\boldsymbol{W}_1 - \boldsymbol{W}_2\|. \tag{64}$$

Hence, $R(\boldsymbol{W})$ is smooth with Lipschitz constant $L_R \le 8\lambda_{\max}$. Since $L_c(\boldsymbol{\theta})$ is also smooth with constant $L_c^{(\text{sm})}$, the overall loss $L(\boldsymbol{\theta})$ is smooth with

$$L^{(\text{sm})} \le L_c^{(\text{sm})} + 8\, \lambda_{\text{reg}} \lambda_{\max}. \tag{65}$$

**Convergence of FEDLAP+.**  By the smoothness of $L(\boldsymbol{\theta})$ and standard results on FEDAVG convergence [33], FEDLAP+ inherits the same convergence guarantees under typical assumptions, i.e.,

$$\mathbb{E}\Big[\|\nabla L(\boldsymbol{\theta}_T)\|^2\Big] = \mathcal{O}\Big(\frac{1}{\sqrt{T}}\Big), \tag{66}$$

where $T$ is the total number of communication rounds.

# E  Additional Results

## E.1  Performance under different partitioning methods

Table 3 presents the node classification accuracy of FEDLAP and FEDLAP+ alongside various previous SFL methods across six benchmark datasets using three partitioning strategies: Louvain, Random, and KMeans. Each experiment involves 10 clients with a 10%–10%–80% train-validation-test split, and results are averaged over 10 independent runs. The Central GNN baseline remains fixed across partitionings, as it is trained on the full graph. Among the partitioning strategies, Louvain generates community-based clusters with fewer inter-client edges, while Random and KMeans typically lead to more fragmented structures and higher inter-client dependencies, making learning more challenging.

As expected, Local GNN suffers most under Random and KMeans partitioning due to missing neighborhood information, especially on datasets with strong structural dependencies like Cora and PubMed. This highlights the importance of collaboration in distributed graph learning.

FEDLAP+ consistently delivers the highest or near-highest accuracy across all datasets and partitioning settings, even under challenging conditions such as Random partitioning on Chameleon or OGBN-Arxiv. Its robustness and strong performance across both high- and low-homophily graphs demonstrate its ability to preserve essential graph information while respecting privacy constraints. This makes FEDLAP+ a practical and reliable solution for real-world SFL applications.

FEDSTRUCT also shows strong performance, particularly under more difficult partitionings, indicating that sharing structural information is effective for learning. However, it lacks privacy guarantees since it requires exchanging graph structure during training. The effectiveness of FEDSTRUCT supports the key idea behind FEDLAP+, which is designed to capture structural signals without directly sharing sensitive graph information.

In contrast, FEDGCN achieves competitive performance but compromises privacy by transmitting aggregated node features (see Appendix C.6.2). FEDSGD and FEDSAGE+ generally underperform, especially under Random and KMeans partitions, highlighting their limitations in leveraging distributed graph structure.

Overall, FEDLAP+ demonstrates a clear advantage by achieving high accuracy across all settings while preserving privacy, establishing it as the most robust and effective method among the compared approaches.

## E.2  Hyperparameters

In the following we provide the hyperparameters used in the experiments, obtained through a grid search to optimize performance. In particular, Table 4 contains, for the different datasets, the learning rate $\lambda$, the weight decay in the L2 regularization, the number of training iterations (epochs), the regularization parameter $\lambda_{\mathrm{reg}}$, the dimensionality of the NSFs, $d_{\mathsf{s}}$, the truncation number $r$, and the model architecture of the node feature and node structure feature predictors, $f_{\boldsymbol{\theta}_{\mathsf{f}}}$ and $g_{\boldsymbol{\theta}_{\mathsf{s}}}$, respectively.

## E.3  Truncation Number Effect

In this experiment, we evaluate the sensitivity of FEDLAP+ to the truncation number $r$, which determines how many eigenvectors of the graph Laplacian are retained in the spectral representation. In Fig. 6, we plot the classification accuracy as a function of $r$ across three datasets, each exhibiting different levels of homophily and structural characteristics:

- Chameleon (left): A heterophilic graph where Laplacian smoothing is typically less effective. We observe that increasing $r$ significantly improves performance, particularly at low $r$, but the gains saturate around $r = 100$. Higher training ratios consistently lead to better accuracy.

- CiteSeer (middle): A moderately homophilic dataset where performance remains relatively stable across a wide range of values of $r$. This indicates that a small number of eigenvectors is sufficient to capture the relevant structural information in this dataset.

Table 3: Node classification accuracy for different partitioning. The results are shown for 10 clients with a 10%–10%–80% train-val-test split. For each result, the mean and standard deviation are shown for 10 independent runs. Edge homophily ratio ($h$) is given in brackets.

| | CORA ($h = 0.81$) | | | CITESEER ($h = 0.74$) | | | PUBMED ($h = 0.80$) | | |
|---|---|---|---|---|---|---|---|---|---|
| CENTRAL GNN | 83.40± 0.63 | | | 70.99± 0.32 | | | 85.60± 0.26 | | |
| | LOUVAIN | RANDOM | KMEANS | LOUVAIN | RANDOM | KMEANS | LOUVAIN | RANDOM | KMEANS |
| FEDSGD GNN | 81.41± 1.24 | 65.26± 1.37 | 67.02± 0.86 | 69.99± 0.91 | 66.53± 1.03 | 67.05± 0.67 | 85.05± 0.32 | 83.96± 0.19 | 84.32± 0.25 |
| FEDSAGE+ | 81.17± 1.26 | 64.53± 1.54 | 66.48± 1.54 | 70.32± 1.06 | 66.57± 0.67 | 67.15± 0.66 | 85.07± 0.32 | 83.97± 0.23 | 84.32± 0.16 |
| FEDPUB | 78.59± 1.31 | 59.17± 1.34 | 61.21± 1.85 | 68.55± 0.85 | 63.30± 1.82 | 63.79± 0.87 | 84.54± 0.22 | 84.00± 0.21 | 83.83± 0.56 |
| FEDGCN-2HOP | 80.82± 1.20 | 82.22± 0.79 | 81.31± 1.07 | 71.25± 0.48 | 71.75± 0.80 | 70.71± 0.64 | 86.10± 0.32 | 86.13± 0.34 | 85.74± 0.24 |
| FEDSTRUCT-P (H2V) | 81.72± 0.84 | 80.01± 1.00 | 79.81± 1.02 | 69.23± 0.91 | 67.51± 1.01 | 68.17± 0.70 | 85.01± 0.29 | 85.40± 0.17 | 85.20± 0.25 |
| FEDLAP | 81.60± 0.79 | 80.55± 0.97 | 80.79± 1.22 | 70.32± 0.58 | 66.29± 0.85 | 67.18± 1.16 | 84.48± 0.34 | 86.43± 0.19 | 85.99± 0.31 |
| FEDLAP+ (ARNOLDI) | 82.01± 0.85 | 79.31± 1.03 | 79.88± 1.16 | 70.07± 0.89 | 67.20± 0.98 | 67.88± 0.83 | 85.16± 0.32 | 85.29± 0.26 | 85.18± 0.31 |
| LOCAL GNN | 75.01± 2.25 | 37.59± 1.12 | 44.95± 3.28 | 59.50± 1.34 | 40.33± 1.20 | 50.27± 6.17 | 81.71± 0.41 | 76.77± 0.25 | 80.31± 0.40 |

| | CHAMELEON ($h = 0.23$) | | | AMAZON PHOTO ($h = 0.82$) | | | OGBN-ARXIV ($h = 0.65$) | | |
|---|---|---|---|---|---|---|---|---|---|
| CENTRAL GNN | 54.38± 1.60 | | | 94.07± 0.41 | | | 68.04± 0.09 | | |
| | LOUVAIN | RANDOM | KMEANS | LOUVAIN | RANDOM | KMEANS | LOUVAIN | RANDOM | KMEANS |
| FEDSGD GNN | 49.02± 1.50 | 35.93± 1.62 | 38.33± 1.25 | 93.60± 0.38 | 89.93± 0.56 | 90.42± 0.43 | 66.70± 0.18 | 54.07± 0.10 | 56.32± 0.11 |
| FEDSAGE+ | 48.60± 1.84 | 35.15± 1.99 | 38.32± 1.24 | 93.52± 0.39 | 89.97± 0.58 | 90.46± 0.34 | ? | ? | ? |
| FEDPUB | 40.44± 1.86 | 34.24± 2.40 | 34.70± 2.10 | 88.74± 1.70 | 88.03± 0.76 | 87.13± 0.99 | 68.50± 0.13 | 55.50± 0.11 | 58.81± 0.12 |
| FEDGCN-2HOP | 49.93± 1.42 | 50.19± 1.34 | 49.97± 1.74 | 93.19± 0.39 | 93.36± 0.44 | 93.62± 0.43 | 65.18± 0.33 | 66.93± 0.14 | 66.20± 0.20 |
| FEDSTRUCT-P (H2V) | 55.72± 1.82 | 55.81± 1.69 | 55.20± 1.43 | 93.73± 0.34 | 92.00± 0.51 | 92.62± 0.39 | 65.62± 0.17 | 64.95± 0.06 | 65.07± 0.23 |
| FEDLAP | 32.81± 2.41 | 32.98± 2.63 | 33.34± 2.37 | 93.28± 0.29 | 92.08± 0.73 | 92.50± 0.45 | 66.73± 0.38 | 66.03± 0.33 | 65.98± 0.33 |
| FEDLAP+ (ARNOLDI) | 54.24± 1.80 | 54.34± 1.59 | 53.95± 1.94 | 93.70± 0.30 | 92.14± 0.56 | 92.53± 0.37 | 68.84± 0.11 | 66.22± 0.26 | 66.56± 0.26 |
| LOCAL GNN | 47.69± 2.20 | 29.53± 1.54 | 30.90± 0.87 | 91.26± 0.57 | 77.62± 0.84 | 78.75± 1.25 | 67.83± 0.14 | 50.43± 0.15 | 55.65± 0.12 |

[2] **FEDGCN lacks privacy** as the server must have access to aggregated node features and 2-hop structures are shared between clients, which constitutes a privacy breach as shown in [16]. Also, the official code overlooks isolated external neighbors removal, potentially enhancing prediction performance above its actual capabilities.

Table 4: Hyper-parameters of the datasets.

| DATA | CORA | CITESEER | PUBMED | CHAMELEON | AMAZON PHOTO | OGBN-ARXIV |
|---|---|---|---|---|---|---|
| $\lambda$ | 0.003 | 0.002 | 0.001 | 0.001 | 0.001 | 0.001 |
| WEIGHT DECAY | 0.0005 | 0.0005 | 0.0003 | 0.0002 | 0.0005 | 0.0001 |
| EPOCHS | 100 | 100 | 150 | 100 | 150 | 1000 |
| $\lambda_{REG}$ | 1 | 1 | 1 | 1 | 0.1 | 1 |
| $d_s$ | 512 | 1024 | 256 | 1024 | 512 | 128 |
| $r$ | 100 | 100 | 100 | 100 | 75 | 75 |
| $\theta_f$ LAYERS | [1433,32, 16, 256,7] | [3703,128,64,64,6] | [500,256,128,64,3] | [2325,256,128,5] | [745,256,8] | [128,128,64, 64,40] |
| $\theta_s$ LAYERS | [512,64,7] | [256,128,64,6] | [256, 64,3] | [1024,5] | [512,128,8] | [1024,40] |

- Amazon Photo (right): A strongly homophilic dataset where accuracy is consistently high, and increasing $r$ yields marginal improvements beyond $r = 50$. The method is more robust to the choice of $r$ in this setting.

These results show that a moderately sized $r$ (e.g., $r = 100$) is sufficient for good performance across a range of datasets and label ratios. Moreover, they validate that spectral truncation effectively reduces model complexity while preserving predictive power, supporting our design of FEDLAP+ for communication-efficient and privacy-preserving SFL.

### E.4 Regularization Coefficient Effect

In Fig. 7, we analyze the sensitivity of FEDLAP and FEDLAP+ to the regularization strength $\lambda_{reg}$, which controls the influence of the Laplacian smoothing term in the optimization objective.

Across all three datasets, we observe that FEDLAP is sensitive to the choice of $\lambda_{reg}$: very small ($\lambda_{reg} = 0$) or very large ($\lambda_{reg} = 100$) values degrade its performance. This behavior reflects under- and over-regularization, respectively. Optimal performance is typically achieved for intermediate values such as $\lambda_{reg} = 1$ or $5$, where structural information is effectively leveraged without overwhelming the learning signal.

In contrast, FEDLAP+ shows remarkable robustness to the choice of $\lambda_{reg}$. Its performance remains relatively stable across a wide range of values. This robustness stems from its spectral truncation mechanism, which implicitly regularizes the model by discarding noisy high-frequency eigenvectors.

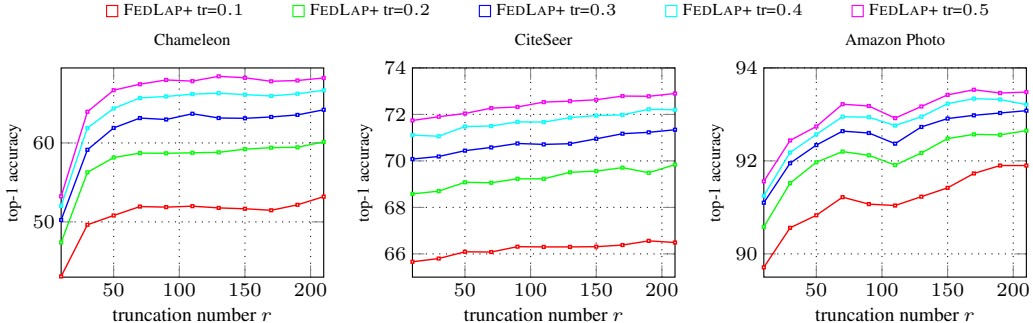

Figure 6: Effect of truncation number $r$ on node classification accuracy for FEDLAP+ across three datasets (Chameleon, CiteSeer, Amazon Photo) under varying training label ratios. Results demonstrate that increasing $r$ generally improves accuracy, with diminishing returns beyond a moderate value (e.g., $r = 100$). Each curve corresponds to a different training ratio $tr \in \{0.1, 0.2, 0.3, 0.4, 0.5\}$.

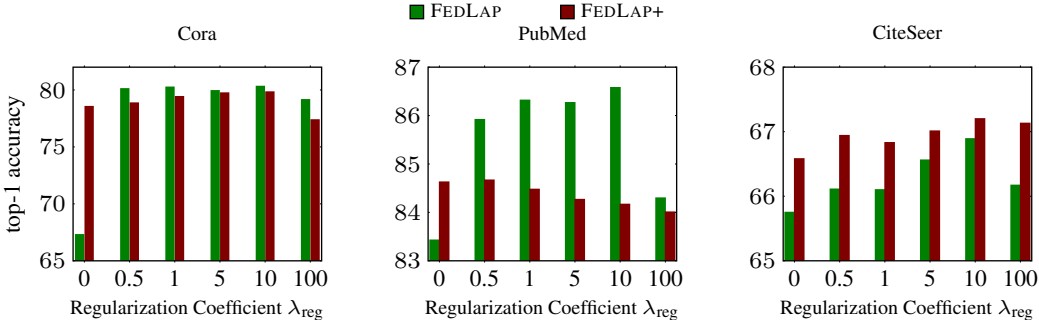

Figure 7: Effect of the regularization coefficient $\lambda_{\text{reg}}$ on node classification accuracy for FEDLAP and FEDLAP+ across three datasets (Cora, PubMed, and CiteSeer). Each bar represents accuracy at a given value of $\lambda_{\text{reg}} \in \{0, 0.5, 1, 5, 10, 100\}$.

As a result, FEDLAP+ benefits less from explicit tuning of $\lambda_{\text{reg}}$, making it a more reliable option in practical scenarios where hyperparameter tuning may be limited or costly.

This robustness further illustrates a key advantage of FEDLAP+: by incorporating structural priors in the spectral domain, it inherently mitigates the need for aggressive regularization, simplifying training and improving stability across diverse datasets.

## F   Communication Cost

Fig. 8 compares several SFL methods across three datasets in terms of accuracy, communication cost, and privacy. The baseline FEDSGD has the lowest communication cost but suffers from low accuracy. FEDGCN offers strong accuracy and low communication cost but lacks privacy, as it directly shares aggregated node features. FEDSTRUCT achieves high accuracy but has poor communication efficiency and does not provide privacy guarantees. FEDSAGE performs poorly in all aspects, with high communication cost, low accuracy, and no privacy protection. In contrast, FEDLAP+ is the only method that performs well across all dimensions—achieving high accuracy, maintaining low communication cost, and preserving privacy—making it the most practical and balanced choice for privacy-sensitive SFL settings.


Figure 8: Comparison of accuracy versus communication cost for different SFL models on three datasets: Chameleon, PubMed, and OGBN-Arxiv. The communication cost is plotted on a logarithmic scale to visualize the variation across several orders of magnitude.

Justification: The abstract and introduction clearly describe the contributions and accurately reflect theoretical and empirical results presented throughout the paper.

Guidelines:

- The answer NA means that the abstract and introduction do not include the claims made in the paper.
- The abstract and/or introduction should clearly state the claims made, including the contributions made in the paper and important assumptions and limitations. A No or NA answer to this question will not be perceived well by the reviewers.
- The claims made should match theoretical and experimental results, and reflect how much the results can be expected to generalize to other settings.
- It is fine to include aspirational goals as motivation as long as it is clear that these goals are not attained by the paper.

2. **Limitations**

Question: Does the paper discuss the limitations of the work performed by the authors?

Answer: [Yes]

Justification: The paper explicitly discusses limitations related to computational complexity and assumptions in the privacy analysis (Sections 5 and 6).

Guidelines:

- The answer NA means that the paper has no limitation while the answer No means that the paper has limitations, but those are not discussed in the paper.
- The authors are encouraged to create a separate "Limitations" section in their paper.
- The paper should point out any strong assumptions and how robust the results are to violations of these assumptions (e.g., independence assumptions, noiseless settings, model well-specification, asymptotic approximations only holding locally). The authors should reflect on how these assumptions might be violated in practice and what the implications would be.
- The authors should reflect on the scope of the claims made, e.g., if the approach was only tested on a few datasets or with a few runs. In general, empirical results often depend on implicit assumptions, which should be articulated.
- The authors should reflect on the factors that influence the performance of the approach. For example, a facial recognition algorithm may perform poorly when image resolution is low or images are taken in low lighting. Or a speech-to-text system might not be used reliably to provide closed captions for online lectures because it fails to handle technical jargon.
- The authors should discuss the computational efficiency of the proposed algorithms and how they scale with dataset size.

- If applicable, the authors should discuss possible limitations of their approach to address problems of privacy and fairness.
- While the authors might fear that complete honesty about limitations might be used by reviewers as grounds for rejection, a worse outcome might be that reviewers discover limitations that aren't acknowledged in the paper. The authors should use their best judgment and recognize that individual actions in favor of transparency play an important role in developing norms that preserve the integrity of the community. Reviewers will be specifically instructed to not penalize honesty concerning limitations.

3. **Theory assumptions and proofs**

   Question: For each theoretical result, does the paper provide the full set of assumptions and a complete (and correct) proof?

   Answer: [Yes]

   Justification: All theoretical results, assumptions, and complete proofs are provided clearly in Sections 4, 5, and detailed in the Appendix.

   Guidelines:

   - The answer NA means that the paper does not include theoretical results.
   - All the theorems, formulas, and proofs in the paper should be numbered and cross-referenced.
   - All assumptions should be clearly stated or referenced in the statement of any theorems.
   - The proofs can either appear in the main paper or the supplemental material, but if they appear in the supplemental material, the authors are encouraged to provide a short proof sketch to provide intuition.
   - Inversely, any informal proof provided in the core of the paper should be complemented by formal proofs provided in appendix or supplemental material.
   - Theorems and Lemmas that the proof relies upon should be properly referenced.

4. **Experimental result reproducibility**

   Question: Does the paper fully disclose all the information needed to reproduce the main experimental results of the paper to the extent that it affects the main claims and/or conclusions of the paper (regardless of whether the code and data are provided or not)?

   Answer: [Yes]

   Justification: The experimental setup, hyperparameters, datasets, and training details necessary for reproducibility are fully documented in Sections 6 and the Appendix.

   Guidelines:

   - The answer NA means that the paper does not include experiments.
   - If the paper includes experiments, a No answer to this question will not be perceived well by the reviewers: Making the paper reproducible is important, regardless of whether the code and data are provided or not.
   - If the contribution is a dataset and/or model, the authors should describe the steps taken to make their results reproducible or verifiable.
   - Depending on the contribution, reproducibility can be accomplished in various ways. For example, if the contribution is a novel architecture, describing the architecture fully might suffice, or if the contribution is a specific model and empirical evaluation, it may be necessary to either make it possible for others to replicate the model with the same dataset, or provide access to the model. In general. releasing code and data is often one good way to accomplish this, but reproducibility can also be provided via detailed instructions for how to replicate the results, access to a hosted model (e.g., in the case of a large language model), releasing of a model checkpoint, or other means that are appropriate to the research performed.
   - While NeurIPS does not require releasing code, the conference does require all submissions to provide some reasonable avenue for reproducibility, which may depend on the nature of the contribution. For example
     (a) If the contribution is primarily a new algorithm, the paper should make it clear how to reproduce that algorithm.

(b) If the contribution is primarily a new model architecture, the paper should describe the architecture clearly and fully.

(c) If the contribution is a new model (e.g., a large language model), then there should either be a way to access this model for reproducing the results or a way to reproduce the model (e.g., with an open-source dataset or instructions for how to construct the dataset).

(d) We recognize that reproducibility may be tricky in some cases, in which case authors are welcome to describe the particular way they provide for reproducibility. In the case of closed-source models, it may be that access to the model is limited in some way (e.g., to registered users), but it should be possible for other researchers to have some path to reproducing or verifying the results.

5. **Open access to data and code**

Question: Does the paper provide open access to the data and code, with sufficient instructions to faithfully reproduce the main experimental results, as described in supplemental material?

Answer: [Yes]

Justification: The code is openly accessible, and a link is provided at the end of the contributions section, including clear instructions for reproducing the experimental results.

Guidelines:

- The answer NA means that paper does not include experiments requiring code.
- Please see the NeurIPS code and data submission guidelines (`https://nips.cc/public/guides/CodeSubmissionPolicy`) for more details.
- While we encourage the release of code and data, we understand that this might not be possible, so "No" is an acceptable answer. Papers cannot be rejected simply for not including code, unless this is central to the contribution (e.g., for a new open-source benchmark).
- The instructions should contain the exact command and environment needed to run to reproduce the results. See the NeurIPS code and data submission guidelines (`https://nips.cc/public/guides/CodeSubmissionPolicy`) for more details.
- The authors should provide instructions on data access and preparation, including how to access the raw data, preprocessed data, intermediate data, and generated data, etc.
- The authors should provide scripts to reproduce all experimental results for the new proposed method and baselines. If only a subset of experiments are reproducible, they should state which ones are omitted from the script and why.
- At submission time, to preserve anonymity, the authors should release anonymized versions (if applicable).
- Providing as much information as possible in supplemental material (appended to the paper) is recommended, but including URLs to data and code is permitted.

6. **Experimental setting/details**

Question: Does the paper specify all the training and test details (e.g., data splits, hyperparameters, how they were chosen, type of optimizer, etc.) necessary to understand the results?

Answer: [Yes]

Justification: All training and testing details, including hyperparameters, data splits, and optimization procedures, are explicitly described in Section 6 and the Appendix.

Guidelines:

- The answer NA means that the paper does not include experiments.
- The experimental setting should be presented in the core of the paper to a level of detail that is necessary to appreciate the results and make sense of them.
- The full details can be provided either with the code, in appendix, or as supplemental material.

7. **Experiment statistical significance**

Question: Does the paper report error bars suitably and correctly defined or other appropriate information about the statistical significance of the experiments?

Answer: [Yes]

Justification: Experimental results include confidence intervals, clearly reporting standard deviations and statistical significance across multiple runs (Tables and Figures in Section 6).

Guidelines:

- The answer NA means that the paper does not include experiments.
- The authors should answer "Yes" if the results are accompanied by error bars, confidence intervals, or statistical significance tests, at least for the experiments that support the main claims of the paper.
- The factors of variability that the error bars are capturing should be clearly stated (for example, train/test split, initialization, random drawing of some parameter, or overall run with given experimental conditions).
- The method for calculating the error bars should be explained (closed form formula, call to a library function, bootstrap, etc.)
- The assumptions made should be given (e.g., Normally distributed errors).
- It should be clear whether the error bar is the standard deviation or the standard error of the mean.
- It is OK to report 1-sigma error bars, but one should state it. The authors should preferably report a 2-sigma error bar than state that they have a 96% CI, if the hypothesis of Normality of errors is not verified.
- For asymmetric distributions, the authors should be careful not to show in tables or figures symmetric error bars that would yield results that are out of range (e.g. negative error rates).
- If error bars are reported in tables or plots, The authors should explain in the text how they were calculated and reference the corresponding figures or tables in the text.

8. **Experiments compute resources**

Question: For each experiment, does the paper provide sufficient information on the computer resources (type of compute workers, memory, time of execution) needed to reproduce the experiments?

Answer: [No]

Justification: The paper currently does not include explicit details about the computational resources required; this will be provided in the supplemental material upon acceptance. The paper introduces a new framework and computational resource details are not crucial for understanding or replicating the main contributions and their impact.

Guidelines:

- The answer NA means that the paper does not include experiments.
- The paper should indicate the type of compute workers CPU or GPU, internal cluster, or cloud provider, including relevant memory and storage.
- The paper should provide the amount of compute required for each of the individual experimental runs as well as estimate the total compute.
- The paper should disclose whether the full research project required more compute than the experiments reported in the paper (e.g., preliminary or failed experiments that didn't make it into the paper).

9. **Code of ethics**

Question: Does the research conducted in the paper conform, in every respect, with the NeurIPS Code of Ethics `https://neurips.cc/public/EthicsGuidelines`?

Answer: [Yes]

Justification: The paper complies fully with the NeurIPS Code of Ethics, involving no ethical concerns or misuse.

Guidelines:

- The answer NA means that the authors have not reviewed the NeurIPS Code of Ethics.

- If the authors answer No, they should explain the special circumstances that require a deviation from the Code of Ethics.
- The authors should make sure to preserve anonymity (e.g., if there is a special consideration due to laws or regulations in their jurisdiction).

10. **Broader impacts**

Question: Does the paper discuss both potential positive societal impacts and negative societal impacts of the work performed?

Answer: [Yes]

Justification: The paper discusses positive societal impacts by enhancing privacy and communication efficiency in federated learning setups. No negative societal impacts were identified.

Guidelines:

- The answer NA means that there is no societal impact of the work performed.
- If the authors answer NA or No, they should explain why their work has no societal impact or why the paper does not address societal impact.
- Examples of negative societal impacts include potential malicious or unintended uses (e.g., disinformation, generating fake profiles, surveillance), fairness considerations (e.g., deployment of technologies that could make decisions that unfairly impact specific groups), privacy considerations, and security considerations.
- The conference expects that many papers will be foundational research and not tied to particular applications, let alone deployments. However, if there is a direct path to any negative applications, the authors should point it out. For example, it is legitimate to point out that an improvement in the quality of generative models could be used to generate deepfakes for disinformation. On the other hand, it is not needed to point out that a generic algorithm for optimizing neural networks could enable people to train models that generate Deepfakes faster.
- The authors should consider possible harms that could arise when the technology is being used as intended and functioning correctly, harms that could arise when the technology is being used as intended but gives incorrect results, and harms following from (intentional or unintentional) misuse of the technology.
- If there are negative societal impacts, the authors could also discuss possible mitigation strategies (e.g., gated release of models, providing defenses in addition to attacks, mechanisms for monitoring misuse, mechanisms to monitor how a system learns from feedback over time, improving the efficiency and accessibility of ML).

11. **Safeguards**

Question: Does the paper describe safeguards that have been put in place for responsible release of data or models that have a high risk for misuse (e.g., pretrained language models, image generators, or scraped datasets)?

Answer: [NA]

Justification: The paper does not involve datasets or models with high risks for misuse.

Guidelines:

- The answer NA means that the paper poses no such risks.
- Released models that have a high risk for misuse or dual-use should be released with necessary safeguards to allow for controlled use of the model, for example by requiring that users adhere to usage guidelines or restrictions to access the model or implementing safety filters.
- Datasets that have been scraped from the Internet could pose safety risks. The authors should describe how they avoided releasing unsafe images.
- We recognize that providing effective safeguards is challenging, and many papers do not require this, but we encourage authors to take this into account and make a best faith effort.

12. **Licenses for existing assets**

Question: Are the creators or original owners of assets (e.g., code, data, models), used in the paper, properly credited and are the license and terms of use explicitly mentioned and properly respected?

Answer: [Yes]

Justification: All datasets and existing models used are clearly cited with references and licenses properly respected, as detailed in Section 6 and the Appendix.

Guidelines:

- The answer NA means that the paper does not use existing assets.
- The authors should cite the original paper that produced the code package or dataset.
- The authors should state which version of the asset is used and, if possible, include a URL.
- The name of the license (e.g., CC-BY 4.0) should be included for each asset.
- For scraped data from a particular source (e.g., website), the copyright and terms of service of that source should be provided.
- If assets are released, the license, copyright information, and terms of use in the package should be provided. For popular datasets, `paperswithcode.com/datasets` has curated licenses for some datasets. Their licensing guide can help determine the license of a dataset.
- For existing datasets that are re-packaged, both the original license and the license of the derived asset (if it has changed) should be provided.
- If this information is not available online, the authors are encouraged to reach out to the asset's creators.

13. **New assets**

    Question: Are new assets introduced in the paper well documented and is the documentation provided alongside the assets?

    Answer: [NA]

    Justification: The paper does not introduce any new datasets, models, or code assets.

    Guidelines:

    - The answer NA means that the paper does not release new assets.
    - Researchers should communicate the details of the dataset/code/model as part of their submissions via structured templates. This includes details about training, license, limitations, etc.
    - The paper should discuss whether and how consent was obtained from people whose asset is used.
    - At submission time, remember to anonymize your assets (if applicable). You can either create an anonymized URL or include an anonymized zip file.

14. **Crowdsourcing and research with human subjects**

    Question: For crowdsourcing experiments and research with human subjects, does the paper include the full text of instructions given to participants and screenshots, if applicable, as well as details about compensation (if any)?

    Answer: [NA]

    Justification: The research presented in this paper does not involve crowdsourcing or human subjects.

    Guidelines:

    - The answer NA means that the paper does not involve crowdsourcing nor research with human subjects.
    - Including this information in the supplemental material is fine, but if the main contribution of the paper involves human subjects, then as much detail as possible should be included in the main paper.
    - According to the NeurIPS Code of Ethics, workers involved in data collection, curation, or other labor should be paid at least the minimum wage in the country of the data collector.

15. **Institutional review board (IRB) approvals or equivalent for research with human subjects**

    Question: Does the paper describe potential risks incurred by study participants, whether such risks were disclosed to the subjects, and whether Institutional Review Board (IRB) approvals (or an equivalent approval/review based on the requirements of your country or institution) were obtained?

    Answer: [NA]

    Justification: The research does not involve human subjects and thus does not require IRB approvals.

    Guidelines:

    - The answer NA means that the paper does not involve crowdsourcing nor research with human subjects.
    - Depending on the country in which research is conducted, IRB approval (or equivalent) may be required for any human subjects research. If you obtained IRB approval, you should clearly state this in the paper.
    - We recognize that the procedures for this may vary significantly between institutions and locations, and we expect authors to adhere to the NeurIPS Code of Ethics and the guidelines for their institution.
    - For initial submissions, do not include any information that would break anonymity (if applicable), such as the institution conducting the review.

16. **Declaration of LLM usage**

    Question: Does the paper describe the usage of LLMs if it is an important, original, or non-standard component of the core methods in this research? Note that if the LLM is used only for writing, editing, or formatting purposes and does not impact the core methodology, scientific rigorousness, or originality of the research, declaration is not required.

    Answer: [NA]

    Justification: The research does not utilize large language models as part of its core methodology.

    Guidelines:

    - The answer NA means that the core method development in this research does not involve LLMs as any important, original, or non-standard components.
    - Please refer to our LLM policy (`https://neurips.cc/Conferences/2025/LLM`) for what should or should not be described.

