# OpenReview forum: "Subgraph Federated Learning via Spectral Methods"
_NeurIPS.cc/2025/Conference — NeurIPS 2025 poster_

### Official Review · Reviewer_Ea2b · 2025-06-21

**Clarity:** 3
**Significance:** 3
**Originality:** 2
**Rating:** 4
**Confidence:** 3

**Summary:**

This paper proposes two methods FEDLAP and FEDLAP+, for Subgraph Federated Learning (SFL) which has gained interest in recent years (some NeurIPS and ICLR papers on this in recent years, for example). In the setting considered in this paper, graph-structured data is partitioned across multiple clients, each with a disjoint subgraph. The central challenge addressed is how to capture global graph structure to improve learning performance without violating privacy or incurring excessive communication costs.

Some key features of the method:
- a graph Laplacian regularizer in the loss function to balance nodes with labels and without labels
- use truncated eigenvectors of the Laplacian to reduce communication and enhance privacy.
- as an offline part of the algorithm, a novel decentralized Arnoldi iteration to compute the spectral components in a privacy-preserving way. A formal privacy analysis is provided for this method.

There is a comprehensive experimental comparison across six benchmark datasets which shows that FEDLAP offers competitive or superior accuracy with better communication efficiency.

**Questions:**

- The privacy analysis assumes a random adjacency matrix, right? More specifically, an adjacency matrix where each element in Bernoulli(p) distributed. Could you extend the analysis to static non-random graphs?

- What does the guarantee exactly tell: are the rest of the edges random and the result protect a specific fixed edge?

- Can you say anything rigorous about the case of finite $n$? Does the privacy analysis require $n \rightarrow \infty $ ?

**Ethical Concerns:**

["NO or VERY MINOR ethics concerns only"]

**Final Justification:**

I find the paper interesting including the membership information based privacy analysis of Thm. 1. However, I have some reservations due to the fact that the current privacy analysis is for the infinite limit case (graph size goes to infinity) and has also some additional assumptions listed in Sec. C.1. Authors promise changes that would address the finite graph case, and I think their suggestions for modifications are convincing but I think that they would require non-trivial changes to the paper, not just cosmetic adjustments. In the current form, the results listed in the main paper are not correct I would say, although with small modifications they can be corrected.

**Limitations:**

yes

**Paper Formatting Concerns:**

nothing

**Quality:**

3

**Strengths And Weaknesses:**

Strengths

- A formal privacy analysis for the offline phase (Theorem 1 and Corollary 1), seems to be the first one for SFL methods (e.g. baselines FEDSTRUCT and FEDGCN lack rigorous privacy proofs).

- significantly reduces communication overhead through the offline sharing of eigenvectors

- The experimental results demonstrate competitive or superior accuracy compared to SOTA methods

Weaknesses:

- The privacy analysis is not entirely rigorous in my opinion: looking at the proofs in the appendix, the results assume randomness of the adjacency matrices. This then gives in the limit those normal distributions stated in Theorem 1. I see here two problems: a) the assumptions of the randomness of $A$ is not clearly stated in the main text and b) the results does not exactly hold for finite $n$.

- It is also not clearly stated what does the privacy guarantee of Thm. 1 mean. It is kind of differential privacy guarantee (privacy loss random variable for edge-DP), however it is not clearly stated.

- No convergence guarantees for the proposed method

---

> ### Author Rebuttal · Authors · 2025-07-31
>
> ## W1: ##
> Our privacy analysis assumes the adjacency matrix  is drawn from a random graph model, where each entry  independently follows a Bernoulli distribution with probability $p$. This assumption, clearly stated in Appendix C.1 and implied in Theorem 1, allows us to analytically derive a closed-form log-likelihood ratio (LLR) and apply the central limit theorem to approximate it with a Gaussian distribution. This approximation facilitates computing the KL divergence, quantifying the adversary's ability to infer edges.
>
> The random graph modeling is standard practice in graph-based privacy analyses, providing controlled analytical insights and clear interpretability [1, 2].
> Importantly, our analysis considers a worst-case attacker with complete spectral knowledge $\mathbf{Q}$, greatly exceeding realistic capabilities, ensuring our theoretical bound remains conservative.
>
> Regarding finite graphs, while the Gaussian approximation is exact only asymptotically, our empirical results (Figure 4) confirm its accuracy for practical, moderately sized graphs. The actual attacker performance observed is consistently weaker than our theoretical bound, reinforcing our claim that the derived guarantees robustly hold even in finite settings.
>
> We will revise the final version of the paper to clearly state the random graph assumption in the main text. Additionally, in our response to your final question, we derive a bound on the error between the true distribution and the Gaussian approximation for finite graphs, further supporting the practical validity of our analysis.
>
> [1] Ullman et al., "Efficiently estimating Erdos-Renyi graphs with node differential privacy," NeurIPS, 2019.
>
> [2] Chen et al., "Private edge density estimation for random graphs: Optimal, efficient and robust," NeurIPS, 2024.
>
> ## W2: ##
> We appreciate the reviewer’s question regarding the privacy guarantee in Theorem 1. We clarify that our privacy analysis is not based on differential privacy (DP), but instead follows a principled approach grounded in membership inference attacks [3].
>
> In our setup, we consider an adversary who observes the spectral projection $\mathbf{U} = \mathbf{A}\mathbf{Q}$, where $\mathbf{Q}$ is a known orthonormal matrix and $\mathbf{A}$ is a random adjacency matrix sampled from a Bernoulli graph model with connection probability $p$. The attacker aims to infer whether a specific edge $A_{uv}$ was present in the computation of $\mathbf{U}$, framing this as a membership inference problem.
>
> To do so, the attacker evaluates the log-likelihood ratio (LLR) between two hypotheses: edge present vs. edge absent. We show analytically that the distributions of the LLR under both hypotheses are nearly indistinguishable for small values of $r$, as measured by the KL divergence.
>
> This analysis aligns with widely-used frameworks in privacy auditing that assess inference risk through optimal hypothesis testing [3]. Using the Neyman–Pearson lemma, we establish that this LLR-based attack is statistically optimal. Moreover, we prove that under practical values of $r, n,$ and $p$, the attacker’s precision and recall satisfy $P + R \leq 1$, which implies that the attacker cannot outperform trivial guessing. Thus, no meaningful structural information can be inferred from $\mathbf{U}$. In other words, a membership inference attack cannot succeed.
>
> We will clarify these concepts in the revised manuscript.
>
> [3] Carlini et al. "Membership inference attacks from first principles." IEEE S&P, 2022.
>
> ## W3: ##
> **Smoothness of the spectral regularizer.**
> Our online loss is defined as:
> \begin{align}
> L(\theta) = L_c(\theta) + \lambda_{\text{reg}} \cdot R(W), \quad \text{where} \quad R(W) = \frac{\operatorname{Tr}(W^\top \Lambda W)}{\operatorname{Tr}(W^\top W)}.
> \end{align}
> Here, $L_c(\theta)$ is the supervised loss (e.g., cross-entropy), $\Lambda$ is the diagonal matrix of Laplacian eigenvalues, and $W$ contains the spectral coefficients. To ensure that $L(\theta)$ is smooth, which is needed for FedAvg convergence, we analyze the smoothness of $R(W)$.
>
> Since $R(W)$ is scale-invariant ($R(\alpha W) = R(W$) for any $\alpha > 0$), we normalize $W$ to unit Frobenius norm ($\lVert W \rVert_{\mathrm{F}} = 1$) after each local update.
>
> On the unit sphere, the gradient becomes:
> \begin{align}
> \nabla_W R(W) = 2(\Lambda W - W \cdot \operatorname{Tr}(W^\top \Lambda W)).
> \end{align}
> This gradient is Lipschitz-continuous. For any $\lVert W_1 \rVert_{\mathrm{F}} = \lVert W_2 \rVert_{\mathrm{F}} = 1$, using $\lVert \Lambda \rVert_2 = \lambda_{\max}$, we have:
> \begin{align}
> \lVert \nabla R(W_1) - \nabla R(W_2) \rVert \le 8 \lambda_{\max} \lVert W_1 - W_2 \rVert
> \end{align}
> So, $R(W)$ is smooth with Lipschitz constant $L_R \le 8 \lambda_{\max}$. Since $L_c(\theta)$ is also smooth, the full loss is smooth with constant:
> $L \le L_c^{(\text{sm})} + 8 \lambda_{\text{reg}} \lambda_{\max}$,
> where $L_c^{(\text{sm})}$ is the Lipschitz constant of $\nabla L_c$.
>
> Thus, FedAvg applied to this loss inherits standard convergence guarantees (e.g., $O(1/\sqrt{T})$ under typical assumptions) [4].
>
> We will include this analysis in the final version of the paper.
>
> [4] Li et al., "On the convergence of fedavg on non-iid data," ICLR, 2020.
>
> ## Q1: ##
>
> Yes, our privacy analysis assumes the adjacency matrix $\mathbf{A}$ is drawn from a random graph model with independent Bernoulli($p$) edges. While real-world graphs are fixed, attackers typically lack structural knowledge, making this prior a natural choice to reflect uncertainty.
>
> This probabilistic model enables a principled privacy analysis using Bayesian inference and privacy auditing. We assess the attacker’s ability to infer whether a specific edge $A_{uv}$ exists based on the spectral observation $\mathbf{U} = \mathbf{A} \mathbf{Q}$, using the log-likelihood ratio and KL divergence to quantify their advantage.
>
> Analyzing a fixed graph would require graph-specific assumptions and limit generality. The random model instead yields conservative, broadly applicable guarantees. Empirically, we observe that attacker performance is even weaker than predicted (Figure 4), confirming the robustness of our approach in static settings.
>
> ## Q2: ##
> To directly address your question, we restate our main result (Theorem 1) in a clear and intuitive way:
>
> Theorem 1 (restated):
> Consider a graph with n nodes, where each edge is included independently with probability $p$. For any pair of nodes u,v, we analyze two hypotheses:
> 1. $H_0$ : u and v are not connected.
> 2. $H_1$ : u and v are connected.
>
> Using spectral methods, we derive an explicit formula for the log-likelihood ratio (LLR) between these hypotheses. The LLR depends on three parameters: the number of nodes n, edge probability p, and the number of shared eigenvectors r.
>
> We then compute the KL divergence between the LLR distributions under $H_0$ and $H_1$, showing it remains small for small r. This implies that an attacker cannot reliably distinguish whether an edge exists based on the spectral representation.
>
> We instantiate a membership inference attack using this LLR: if $\text{LLR}_{uv} > \gamma$, the attacker predicts an edge; otherwise, no edge. We analytically evaluate the attack’s Precision (P) and Recall (R), showing that $P + R \leq 1$. This means the attacker performs worse than trivial baselines (e.g., guessing all edges exist or none do).
>
> Therefore, even an optimal attacker gains no meaningful advantage from observing the spectral matrix $\mathbf{U}$, and individual edge privacy is effectively preserved.
>
> ## Q3: ##
> The privacy analysis presented in our paper utilizes a central limit theorem (CLT) to approximate the LLR as a Gaussian distribution.
> To address the reviewer’s concern regarding finite-sample effects, we invoke the multivariate Berry–Esseen theorem [5], a standard tool for quantifying the convergence rate of the central limit theorem. Specifically, using Equation (38) from the paper, we construct a zero-mean random vector $\mathbf{x}$ with normalized covariance, formed from the Bernoulli(p) adjacency entries $A_{ij}$ and the orthonormal spectral matrix $\mathbf{Q}$:
>
> \begin{align}
> \\mathbf{x} = \\frac{1}{\\sqrt{n p (1-p)}} (\\mathbf{A}_{u,:} - p\\mathbf{1})^\\top \\mathbf{Q}
> \end{align}
>
> This normalization ensures that $\mathbf{x}$ has zero mean and identity covariance. The Berry–Essen bound then quantifies how closely the distribution of $\mathbf{x}$ approximates a multivariate normal distribution, even for finite n.
> Since each entry in $ \mathbf{A}_{u,:}$ is Bernoulli(p) (with typically small p), the Berry–Esseen bound states that the maximum deviation from Gaussian distribution in terms of cumulative distribution functions scales as $O\left(\frac{1}{\sqrt{n p (1-p)}}\right)$. When $p$ is small, this simplifies approximately to:
>
> \begin{align}
> \\text{Error}_{\\text{CLT}} = O\\left(\\frac{1}{\\sqrt{n p}}\\right).
> \\end{align}
>
> This result clearly indicates the number of samples $n$ needed for the CLT approximation to hold rigorously: specifically, the error decreases rapidly as $np$ grows. Therefore, even moderate-size graphs yield accurate approximations due to the typical sparsity of real-world graphs (small $p$) and reasonably large $n$.
>
> Importantly, we stress that the assumption of an infinite sample (or large n) is only used to simplify the analytical derivation of the LLR and ensure the CLT approximation’s validity. It does not inherently result in weaker privacy guarantees for smaller graphs. Indeed, as shown empirically in our experiments (see Fig. 4), even when the CLT approximation is not exact, the attacker’s real-world performance remains weaker than our theoretical analysis predicts. Thus, our theoretical bound remains conservative, and our privacy guarantees remain robust even for finite graphs.
>
> [5] Friedrich, "A Berry-Esseen bound for functions of independent random variables," The Annals of Statistics, 1989.

---

> > ### Comment · Reviewer_Ea2b · 2025-08-03
> >
> > Thank you for the comprehensive rebuttal. I would recommend clarifying the connection of Thm. 1 to those prior membership inference works as you describe, to put it better into context. I recommend adding a finite dimensional version of Thm. 1 via Berry-Esseen, as you describe.

---

> > > ### Author Response · Authors · 2025-08-04
> > >
> > > Thank you for your comment and for finding our rebuttal helpful. We agree with your suggestions, which we believe will further strengthen the paper.
> > >
> > > Following your advice, we will clarify the connection between Theorem 1 and membership inference attacks to better contextualize our contribution in the revised version of the manuscript. We will also include a finite-dimensional version of Theorem 1 using the Berry-Esseen theorem, as you recommended.
> > > If there are no further concerns and our response has addressed your comments satisfactorily, we would be grateful if you would consider updating your score accordingly.

---

### Official Review · Reviewer_7gcZ · 2025-07-03

**Clarity:** 3
**Significance:** 2
**Originality:** 3
**Rating:** 4
**Confidence:** 3

**Summary:**

The authors propose a method which is able to cater for graph-structured data in the federated learning domain. It claims that it improves on the state of the art on both communication and overall performance.

**Questions:**

Based on my prior comments I would like the authors to answer the following questions:

- Can you please expand a bit more on the privacy? Normally, such methods require a form of statistical privacy (e.g. DP) to avoid data leaks and inference attacks. At the very least some sort of formal encryption or similar. As presented, I currently lack understanding on why the definition provided can be considered a form of "strong" or "formal" privacy definition.
- Can you please elaborate a bit in terms of the hardware resources required? I feel it is quite important given the context of the paper.

**Ethical Concerns:**

["NO or VERY MINOR ethics concerns only"]

**Final Justification:**

After reading the authors comments, I am willing to raise my score to borderline accept given the justification provided. The paper does not appear to have any technical flaws but I still I am highly concerned about the definition of privacy used (and the protections it entails).

I would highly encourage the authors to substantially rephrase and improve that aspect within the paper so that the limitations are clear and explicit.

**Limitations:**

yes

**Paper Formatting Concerns:**

Nothing of note.

**Quality:**

2

**Strengths And Weaknesses:**

The paper tackles an important and unsolved problem: how to efficiently train in the FL domain when using graph-structured data, which is an important use-case.

The paper strengths can be summarised as follows:

- The performance is appealing and is compared against a number of competing methods
- The code is provided for the reviewing process
- The evaluation is extensive in terms of overall results regarding performance

However the paper suffers from severe flaws which I am outlining below:

- Claims of strong / formal privacy -- to me this is the biggest problem with this paper. Normally, privacy definitions are quite strict and the notion of "not being able to infer the connections of nodes" is not a form of _strong_ privacy or any formal definition of it. I would argue the method provides _no_ privacy at all apart from being unable to infer who your neighbors are.
- Little light on the details regarding experimentation hardware used and potential scalability of the method (although results are extensive in terms of numbers).

---

> ### Author Rebuttal · Authors · 2025-07-31
>
> We believe we have addressed all of your comments thoroughly and satisfactorily. If our responses resolve your concerns, we would kindly ask you to consider updating your score accordingly.
>
> ## Weakness 1. ##
> * Claims of strong/formal privacy -- to me, this is the biggest problem with this paper. Normally, privacy definitions are quite strict and the notion of "not being able to infer the connections of nodes" is not a form of strong privacy or any formal definition of it. I would argue the method provides no privacy at all, apart from being unable to infer who your neighbors are.
>
> ## Question 1. ##
>
> * Can you please expand a bit more on the privacy? Normally, such methods require a form of statistical privacy (e.g. DP) to avoid data leaks and inference attacks. At the very least some sort of formal encryption or similar. As presented, I currently lack understanding on why the definition provided can be considered a form of "strong" or "formal" privacy definition.
>
> ## Response. ##
> Our privacy measure naturally aligns with a membership inference formulation, widely adopted in privacy auditing [1,2]. Below, we justify our privacy analysis, why our results imply strong privacy guarantees, and explain why differential privacy is not well suited to our setting.
>
> **Privacy in FEDLAP+: What we protect and why it matters**
> To properly address the concerns raised regarding the privacy analysis, we first clarify how FedLAP+ approaches the Subgraph Federated Learning (SFL) problem. FedLAP+ is designed as a general privacy-preserving framework that leverage structural information (such as connectivity patterns and degree statistics) among clients in a decentralized setting. As explained in the paper, FedLAP+ separates the SFL problem into two distinct sub-problems (phases):
>
> 1. Offline Phase:
> - This phase is executed once, prior to training.
> - No node features or sensitive node attributes are exchanged in this phase; only information related to the graph structure and interconnections is shared.
> - The primary question FedLAP+ addresses here is how to efficiently extract useful structural information from interconnections while preserving privacy. By working in the spectral domain, we demonstrate experimentally that most valuable information regarding interconnections resides in a small number of dominant eigenvectors of the graph Laplacian.
>
>
> 2. Online Phase:
> - This phase is identical to standard federated learning (FL) training process.
> - No additional information beyond model parameters is exchanged between clients and the server. Thus, FedLAP+’s online phase inherits all privacy guarantees and protections inherent in standard FL.
> - Consequently, any privacy-enhancing methods used in standatd FL (e.g., differential privacy or secure aggregation)  can be applied directly to FedLAP+’s online phase.
>
> Given this decomposition, ensuring the privacy of FedLAP+ reduces solely to analyzing the offline phase, since the online phase does not introduce additional privacy leakage beyond that of standard FL.
>
> In the offline phase, the only possible privacy leakage relates to node connections,
> since no sensitive node features are exchanged in this phase. Hence, our analysis explicitly addresses privacy concerning these structural connections.
>
> **Clarification on Privacy Analysis (Offline Phase)**
>
> To clarify precisely how our privacy analysis works, we first explain the exact setup:
>
> We assume a binary random adjacency matrix $\mathbf{A}$ with probability of connection $p$. To rigorously assess privacy, we consider a worst-case scenario where the attacker possesses complete spectral knowledge, far beyond realistic capabilities. Given the relation $\mathbf{U} = \mathbf{A} \mathbf{Q}$, where $\mathbf{Q} \in \mathbb{R}^{n \times r}$ is a known orthonormal matrix and $\mathbf{U} \in \mathbb{R}^{n \times r}$, the attacker observes both $\mathbf{U}$ and $\mathbf{Q}$ and attempts to infer $\mathbf{A}$. It is crucial to note that this process is entirely deterministic. Crucially, the attacker cannot recover $\mathbf{A}$ via simple inversion, $\mathbf{A} = \mathbf{U}\mathbf{Q}^{-1}$,  since the problem is underdetermined due to the low-rank nature of the matrix $\mathbf{A}$. There exist infinitely many adjacency matrices consistent with a given $\mathbf{U}$ and $\mathbf{Q}$.  Moreover, we also assume that the attacker knows $p$, hence that $\boldsymbol{A}$ is sparse (i.e., most nodes are not connected).
>
>
> In this setup, differential privacy (DP) is not a suitable privacy metric. DP measures the indistinguishability between two neighboring datasets differing in a single element under a randomized mechanism. However, since our scenario is deterministic, an attacker can trivially distinguish between two adjacency matrices $\mathbf{A}$ and $\mathbf{A}’$ that differ in a single edge by simply checking whether they satisfy $\mathbf{U} = \mathbf{A}\mathbf{Q}$.
>
> Instead, we adopt a membership inference-based approach, which is widely used in empirical privacy auditing [1,2]. Specifically, we consider an attacker who attempts to determine whether a specific connection $A_{uv}$ was involved in computing $\mathbf{U}$. This leads to a binary hypothesis testing problem, where the attacker computes the log-likelihood ratio (LLR) of observing $\mathbf{U}$ under the two hypotheses (with and without the presence of $A_{uv}$). Following the Neyman–Pearson lemma, this LLR test is the optimal decision rule.
>
> Our analysis shows that even under this optimal attack, for practical values of parameters $r$, $n$, and $p$,  the attacker performs no better than random guessing (i.e., $\text{Precision} (P) + \text{Recall} (R) \leq 1$). Practically, the attacker cannot extract any meaningful structural information from observing the spectral matrix $\mathbf{U}$.
>
> Additionally, we derive an explicit analytical formula for the KL divergence between the distributions under the null hypothesis (edge absent) and the alternative hypothesis (edge present) and show that, for small values of $r$, it is extremely small, indicating the two distributions are nearly indistinguishable. Practically, this implies that even an optimal attacker would have negligible advantage in accurately identifying specific edges. In other words, a membership inference attack cannot succeed.
>
> Regarding the reviewer’s claim that our privacy definition and guarantees provide "no privacy at all apart from being unable to infer who your neighbors are," we respectfully point out that this is precisely the privacy risk of FEDLAP+ (as discussed above under the heading "Privacy in FEDLAP+: What we protect and why it matters"), and the one we rigorously address.
>
>
> [1] Carlini et al. "Membership inference attacks from first principles." SP, 2022.
>
> [2] Zarifzadeh et al., "Low-cost high-power membership inference attacks." ICML, 2024.
>
> ## Weakness 2. ##
> Little light on the details regarding experimentation hardware used and potential scalability of the method (although results are extensive in terms of numbers).
>
> ## Question 2. ##
> Can you please elaborate a bit in terms of the hardware resources required? I feel it is quite important given the context of the paper.
>
> ## Response.  ##
> Our experiments were conducted on a machine with 2 × NVIDIA Tesla V100 SXM2 GPUs, each with 32GB of RAM, connected via NVLink.
>
> As discussed in the paper, the online phase of FedLAP is both computationally and communication-wise equivalent to standard federated learning (FL). Therefore, it inherits the same scalability and efficiency properties as classical FL frameworks.
>
> The offline phase, which involves the decentralized Arnoldi iteration, is analyzed in detail in the paper (Appendix B.3). Its computation and communication complexity grows linearly with the number of nodes in the graph, making it practical even for large graphs. Moreover, because real-world graphs typically have sparse adjacency matrices, the memory footprint for graph-related computations is small, and the overall procedure can be executed efficiently.
>
> We will include a more detailed description of the hardware setup in the revised manuscript.
>
> ## Final comment ##
> We respectfully note that a score of 2 is typically intended for papers with technical flaws, weak evaluation, inadequate reproducibility, or incompletely addressed ethical considerations. We appreciate the reviewer’s concern regarding the framing of our privacy claims and have addressed this point in our response. However, we believe this issue relates to only one aspect of the work. Our paper presents a broader framework with multiple contributions, and we feel that both the review and the overall score should reflect the work as a whole. We kindly ask that this be taken into consideration.

---

> > ### Comment · Reviewer_7gcZ · 2025-08-02
> > **Read your rebuttal.**
> >
> > Thanks for this, please see my final justification comments. I agree with certain parts of your rebuttal and raised my score accordingly, but still my concerns remain.

---

> > > ### Author Response · Authors · 2025-08-02
> > >
> > > Thank you for reading our rebuttal. We appreciate that you found parts of it helpful.
> > > As of now, we do not have access to your final justification, since this becomes visible only after the decision is released, in line with the NeurIPS process.
> > >
> > > To follow the NeurIPS guidelines encouraging "engaging in an open exchange with the authors" during this phase, we would be grateful if you could share any remaining concerns or clarifications directly with us. We would be glad to engage further and do our best to address your remaining concerns.
> > >
> > > Thank you again for your time and effort in the review process.

---

> > > > ### Author Response · Authors · 2025-08-06
> > > >
> > > > Dear Reviewer,
> > > >
> > > > we would greatly appreciate it if you could share any remaining concerns you might have, as we do not have access to your final justification. We’d be happy to clarify or address any remaining points.
> > > >
> > > > Thank you again for your time.
> > > >
> > > > Sincerely,
> > > >
> > > > The Authors

---

### Official Review · Reviewer_3cQg · 2025-07-03

**Clarity:** 3
**Significance:** 3
**Originality:** 3
**Rating:** 4
**Confidence:** 3

**Summary:**

This paper presents  framework, FEDLAP,  for subgraph federated learning that addresses the challenge of learning from graph-structured data distributed across multiple clients while preserving privacy.

The key contribution lies in leveraging Laplacian smoothing in the spectral domain to capture inter-node dependencies without requiring the exchange of sensitive node embeddings or computationally expensive structural information sharing that existing methods rely on.

The authors propose two variants: FEDLAP, which uses Laplacian regularization to implicitly enforce structural similarity among neighboring nodes, and FEDLAP+, which operates in the spectral domain using truncated eigendecomposition to reduce communication overhead and enhance privacy.

**Questions:**

No more questions, please check the weakness.

**Ethical Concerns:**

["NO or VERY MINOR ethics concerns only"]

**Limitations:**

yes

**Quality:**

3

**Strengths And Weaknesses:**

**Strengths:** This paper makes several significant contributions to subgraph federated learning that address real limitations in existing work. The theoretical foundation is solid, with FEDLAP being the first SFL method to provide formal privacy guarantees through rigorous analysis of potential attacks using log-likelihood ratios and KL divergence bounds.

**Weaknesses:** Despite its strengths, the paper has several limitations that could impact its practical adoption. The privacy analysis, while theoretically sound, relies on strong assumptions (known connection probabilities, access to spectral matrices) that may not hold in practice, and the actual attack performance shown in Figure 4 is weaker than theoretical predictions, suggesting the bounds may be loose. The method's performance is inconsistent across graph types—while FEDLAP excels on homophilic graphs, it significantly underperforms on heterophilic graphs like Chameleon, limiting its general applicability.

The decentralized Arnoldi iteration requires homomorphic encryption, which adds computational overhead and implementation complexity that isn't fully characterized in terms of practical costs. The communication analysis, while showing theoretical improvements, doesn't account for the encryption overhead or provide empirical measurements of actual network traffic.

Additionally, the truncation parameter r requires careful tuning and domain knowledge about graph structure (number of communities), which may not be readily available in practice.

The experimental setup is somewhat limited, focusing primarily on semi-supervised node classification with only 10% labeled nodes, and doesn't explore other important graph learning tasks or more challenging federated settings with severe data heterogeneity.

---

> ### Author Rebuttal · Authors · 2025-07-31
>
> We thank the reviewer for the thoughtful and detailed feedback, which has helped us improve the paper. We believe we have addressed all of your comments thoroughly and satisfactorily. If our responses resolve your concerns, we would be grateful if you would consider updating your score accordingly.
>
> # WEAKNESSES: #
>
> ## W1: ##
> We thank the reviewer for the comment. In our privacy analysis of FEDLAP+, we deliberately consider a strong attacker model during the offline phase of the protocol. Specifically, we assume that the attacker (a malicious client) has access to:
> 1. the spectral projection matrix $Q_{\mathcal{V}_1,:}$, and
> 2. the connection probability $p$,
>
> even though these are not shared in practice.
>
> We emphasize that this analysis is intentionally conservative. The goal is to establish a worst-case privacy guarantee, following the standard practice in the privacy literature (e.g., in secure aggregation or spectral privacy frameworks), where one assumes an unrealistically strong attacker. This approach is crucial for stress-testing the protocol and identifying fundamental vulnerabilities. If FEDLAP+ is private under this strong model, it is by design private under weaker, more realistic threat models.
>
> By showing that even under this pessimistic scenario the attacker gains negligible advantage (e.g., precision + recall < 1), we provide strong assurance that the offline phase of FEDLAP+ preserves link privacy. In particular, this result implies that even if spectral components were somehow leaked, the attacker still cannot reliably infer internal connections of other clients.
>
> This level of robustness is especially valuable in federated environments, where partial leakage or side-channel access may occur. Our analysis thus gives clients confidence that such leakage does not compromise their local subgraph structure.
>
> We hope this clarifies the rationale behind our assumptions: they are not intended to reflect typical attacker capabilities, but rather to provide rigorous evidence that FEDLAP+ is fundamentally privacy-preserving.
>
> Regarding Fig. 4, the fact that the real attack performs worse than the theoretical bound is expected (at best, the real attack can perform as the bound). This gap further strengthens our privacy claims. It shows that our bound is conservative, reinforcing the privacy guarantees of FEDLAP+.
>
> ## W2: ##
> We appreciate the reviewer’s observation. In this paper, our primary contribution is FEDLAP+, a spectral-domain method designed to handle both homophilic and heterophilic graph structures. FEDLAP+ builds on a base variant we call FEDLAP, which applies Laplacian smoothing directly in the spatial domain.
>
> As highlighted in line 323 of the main paper, the base FEDLAP model relies on the assumption that neighboring nodes tend to have similar features or labels, an assumption that aligns well with homophilic graphs. However, in heterophilic graphs such as Chameleon, where connected nodes often belong to different classes or have dissimilar features, this assumption no longer holds. As a result, Laplacian smoothing can blur useful distinctions between nodes and degrade performance. This is a well-known limitation of smoothing-based methods and is expected in our setting as well.
>
> Importantly, FEDLAP was introduced primarily to motivate FEDLAP+, and FEDLAP's results are included to show that it performs well in the homophilic case, often on par with existing state-of-the-art methods. However, we do not position FEDLAP as a standalone solution for heterophilic graphs. Instead, FEDLAP+ addresses this limitation by operating in the spectral domain, where it can selectively retain global structure and filter out noisy or misleading local signals. FEDLAP+ achieves excellent performance for heterophilic graphs such as Chameleon.
>
> ## W3: ##
> We thank the reviewer for highlighting this practical concern. To clarify, we use Homomorphic Encryption (HE) only once, during the offline decentralized Arnoldi iteration prior to training. Its purpose is to securely aggregate the spectral information (specifically the projection matrix $\mathbf{Q})$ without revealing individual node features or adjacency details.
>
> While HE does introduce extra computational and communication overhead, this cost is fixed per iteration and does not scale unfavourably with the number of nodes or clients. As discussed in Section E.2 of the FedGCN paper [1], standard HE methods such as CKKS typically increase communication costs by roughly 15x compared to plaintext. However, by applying optimizations like Boolean packing, this overhead can be significantly reduced, down to roughly twice that of plaintext communication for large vectors.
>
> Importantly, our privacy analysis assumes a worst-case scenario where the entire spectral matrix $\mathbf{Q}$ is directly revealed to the attacker. Therefore, even without relying on HE, our analysis already demonstrates strong privacy guarantees. The use of HE thus serves as an additional practical layer of security, but our privacy guarantees hold true regardless of its usage.
>
> We will include the cost of the HE in the analysis of the communication in the revised version of the paper. However, we emphasize that HE is not a requirement for FEDLAP+ to ensure privacy.
>
> [1] Yao et al., "FedGCN: Convergence-communication tradeoffs in federated training of graph convolutional networks," NeurIPS, 2023.
>
> ## W4: ##
> We thank the reviewer for pointing out the importance of the spectral truncation parameter $r$ in FEDLAP+. This parameter controls how many eigenvectors of the graph Laplacian are retained during the spectral decomposition step.
>
> More precisely, the global Laplacian matrix $L_G$ is approximated by the low-rank decomposition $L_G \approx U_{[:, r]} \Lambda_{[r, r]} U_{[:, r]}^\top$, where $U_{[:, r]} \in \mathbb{R}^{n \times r}$ contains the first $r$ eigenvectors (corresponding to the smallest eigenvalues), and $\Lambda_{[r, r]}$ is the diagonal matrix of the corresponding eigenvalues. The matrix $W \in \mathbb{R}^{r \times d_s}$ is then learned in the spectral domain. Truncating to the top-$r$ smooth eigenvectors serves three purposes:
> - It reduces communication (only the top $r$ eigenvectors are shared),
> - it improves privacy (smaller $r$ reduces potential leakage),
> - It acts as a regularizer, filtering out high-frequency components (which often capture noise) that may harm generalization.
>
> To choose $r$ in practice:
> - A rule of thumb is to set $r$ close to the intrinsic rank of the graph, which often corresponds to the number of clusters or communities. For many real-world datasets, this number is much smaller than the total number of nodes $n$.
> - In our experiments (see Fig.5 and Fig.6 of the paper), we observed that moderate values of $r$ (e.g., 50–200) work well across all datasets. For example, in PubMed (with over 19K nodes), $r = 80$ already gives strong performance, and further increasing $r$ yields diminishing returns.
> - We also show in Fig.5 (left panel) and Fig.6 that the performance of FEDLAP+ is robust across a wide range of $r$ values, especially when the training data is limited, indicating that fine-tuning $r$ is not critical.
>
> We will include this detailed explanation of the truncation parameter $r$ in the revised manuscript.
>
> ## W5: ##
> We would like to clarify that our primary objective in designing FEDLAP+ was to develop a privacy-preserving framework that incorporates the knowledge of interconnections in a decentralized setting, while keeping the setup as general and aligned with standard Federated Learning (FL) as possible. To this end, the online phase of FEDLAP is identical to standard FL, ensuring that any specific FL scenario can be readily integrated into our framework.
>
> Regarding our experiments, we intentionally evaluated FEDLAP+ across multiple representative federated setups to cover varying degrees of client heterogeneity and interconnections. Specifically, we adopted three distinct partitioning strategies (Random, K-means, and Louvain) each corresponding to a unique FL scenario, as detailed in Tables 2 and 3 of our paper:
> 1. Random partitioning: Nodes are randomly distributed across clients, resulting in a large number of interconnections and relatively homogeneous clients.
> 2. K-means partitioning: Nodes with similar features are clustered together, creating highly heterogeneous clients with fewer interconnections.
> 3. Louvain partitioning: Nodes within the same community are grouped into the same client, leading to high client heterogeneity and minimal interconnections.
>
> These setups allow us to analyze FEDLAP+’s behavior under a range of realistic and challenging federated settings.
>
> Additionally, in Fig. 5, we reported the performance of FEDLAP+ with varying numbers of clients (i.e., small and large node counts per client) and different training ratios, further enriching the diversity of experiments.
>
> As for our focus on node classification, our primary focus in this paper was to demonstrate that FedLAP+ achieves strong privacy guarantees while maintaining competitive utility. As the majority of existing SFL literature focuses on node classification, we adopt the same setting to enable fair and meaningful comparisons with prior work.
>
> That said, FedLAP+ can be easily extended to other tasks, such as edge classification. We will include result for such tasks in the final version of the paper to further demonstrate the framework’s versatility.
>
> We would appreciate it if the reviewer could specify any particular FL scenario or experimental setup they believe would be valuable to investigate.

---

> > ### Comment · Reviewer_3cQg · 2025-08-05
> >
> > Thank you for your response. My concern has been largely resolved, and I will maintain my original rating score.

---

> > > ### Author Response · Authors · 2025-08-06
> > >
> > > Thank you for your follow-up and for taking the time to review our response. We're glad to hear that your concerns have been largely resolved.

---

### Official Review · Reviewer_HaMS · 2025-07-03

**Clarity:** 3
**Significance:** 3
**Originality:** 2
**Rating:** 4
**Confidence:** 4

**Summary:**

This paper proposes FEDLAP and its spectral variant FEDLAP+, novel frameworks for subgraph federated learning that leverage Laplacian smoothing in both spatial and spectral domains to capture global graph structure while preserving privacy. By avoiding the exchange of sensitive node features and using a decentralized Arnoldi iteration for efficient spectral decomposition, the approach reduces communication overhead and enhances scalability. The authors provide formal privacy guarantees and demonstrate through extensive experiments that FEDLAP achieves competitive or superior accuracy compared to existing SFL methods, especially on large or heterophilic graphs.

**Questions:**

1. The privacy guarantees rely on strong assumptions, such as attackers having access to certain spectral components. Could the authors elaborate on how realistic these assumptions are in practice?
2. While FEDLAP+ improves performance on heterophilic graphs, the base FEDLAP model performs poorly in such settings. Could the authors provide more insight into this limitation?
3. The spectral truncation parameter  𝑟 plays a key role in FEDLAP+.  Can the authors provide more guidance on how to choose it across different datasets?
4. In real-world federated settings, the graph structure may evolve over time. How adaptable is FEDLAP/FEDLAP+ to dynamic graphs? Can the spectral components be incrementally updated, or must the entire process be rerun?
5. On Chameleon, FEDSTRUCT outperforms FEDLAP. Could the authors explain the limitations of Laplacian smoothing in these cases, and whether hybrid approaches (combining spectral and feature-level smoothing) might help?

**Ethical Concerns:**

["NO or VERY MINOR ethics concerns only"]

**Limitations:**

YES

**Quality:**

3

**Strengths And Weaknesses:**

1. This paper presents a novel, privacy-preserving approach to subgraph federated learning using Laplacian smoothing and spectral methods.  It offers strong theoretical grounding, including formal privacy guarantees, and demonstrates competitive or superior performance across multiple datasets.  The proposed method reduces communication overhead and scales well.
2. The privacy analysis relies on idealized assumptions, and the method is mainly tested on node classification, limiting its generality. FEDLAP underperforms on heterophilic graphs, and the computational cost of spectral decomposition is not fully discussed.

---

> ### Author Rebuttal · Authors · 2025-07-30
>
> We thank the reviewer for the thoughtful and detailed feedback, which has helped us improve the paper. We believe we have addressed all of your comments thoroughly and satisfactorily. If our responses resolve your concerns, we would be grateful if you would consider updating your score accordingly.
>
> ## WEAKNESSES: ##
>
> **Note**: We address the idealized assumptions in the response to your Question 1 and the comment on heterophilic graphs in the response to your Question 2.
>
> **Generality beyond node classification.** FedLAP+ is designed as a general privacy-preserving framework that leverages structural information (such as connectivity patterns and degree statistics) across clients in a decentralized setting. It naturally decomposes the SFL problem into two phases:
> - Offline Phase: Executed once before training, this phase privately extracts useful graph-level structure without sharing node features or labels.
> - Online Phase: Identical to standard FL, this phase supports any local graph learning task, including node classification, edge prediction, or link-level inference.
>
> Our primary focus in this paper was to demonstrate that FedLAP+ achieves strong privacy guarantees while maintaining competitive utility. As the majority of existing SFL literature focuses on node classification, we adopt the same setting to enable fair and meaningful comparisons with prior work.
>
> That said, FedLAP+ can be easily extended to other tasks, such as edge classification. We will include results for such tasks in the final version of the paper to further demonstrate the framework’s versatility.
>
> **Computational cost of spectral decomposition:**
> The spectral decomposition in FedLAP+ is performed during the offline phase using the Arnoldi iteration, a well-known method for efficiently computing a few dominant eigenvectors of large sparse matrices.
>
> We explicitly discuss the computational cost of the Arnoldi iteration at the beginning of Section 4.3 and provide a more detailed analysis in Appendix B.3. Importantly, the computational complexity of Arnoldi iteration scales linearly with the number of nodes in the graph, making it well suited for large and sparse graphs. This enables fast and scalable spectral decomposition, even for sizable real-world datasets.
>
> ## QUESTIONS: ##
>
> ## Q1: ##
> We thank the reviewer for the comment. In our privacy analysis of FEDLAP+, we deliberately consider a strong attacker model during the offline phase of the protocol. Specifically, we assume that the attacker (a malicious client) has access to:
> (i) the spectral projection matrix $Q_{\mathcal{V}_1,:}$, and
> (ii) the connection probability $p$,
> Even though these are not shared in practice.
>
> We emphasize that this analysis is intentionally conservative. The goal is to establish a worst-case privacy guarantee, following the standard practice in the privacy literature (e.g., in secure aggregation or spectral privacy frameworks), where one assumes an unrealistically strong attacker. This approach is crucial for stress-testing the protocol and identifying fundamental vulnerabilities. If FEDLAP+ is private under this strong model, it is by design private under weaker, more realistic threat models.
>
> By showing that even under this pessimistic scenario the attacker gains negligible advantage (e.g., precision + recall < 1), we provide strong assurance that the offline phase of FEDLAP+ preserves link privacy. In particular, this result implies that even if spectral components were somehow leaked, the attacker still cannot reliably infer internal connections of other clients.
>
> This level of robustness is especially valuable in federated environments, where partial leakage or side-channel access may occur. Our analysis thus gives clients confidence that such leakage does not compromise their local subgraph structure.
>
> We hope this clarifies the rationale behind our assumptions: they are not intended to reflect typical attacker capabilities, but rather to provide rigorous evidence that FEDLAP+ is fundamentally privacy-preserving.
>
> ## Q2: ##
> We appreciate the reviewer’s observation. In this paper, our primary contribution is FEDLAP+, a spectral-domain method designed to handle both homophilic and heterophilic graph structures. FEDLAP+ builds on a base variant we call FEDLAP, which applies Laplacian smoothing directly in the spatial domain.
>
> As highlighted in line 323 of the main paper, the base FEDLAP model relies on the assumption that neighboring nodes tend to have similar features or labels, an assumption that aligns well with homophilic graphs. However, in heterophilic graphs, where connected nodes often belong to different classes or have dissimilar features, this assumption no longer holds. As a result, Laplacian smoothing can blur useful distinctions between nodes and degrade performance. This is a well-known limitation of smoothing-based methods and is expected in our setting as well.
>
> Importantly, FEDLAP was introduced primarily to motivate FEDLAP+, and FEDLAP's results are included to show that it performs well in the homophilic case, often on par with existing state-of-the-art methods. However, we do not position FEDLAP as a standalone solution for heterophilic graphs. Instead, FEDLAP+ addresses this limitation by operating in the spectral domain, where it can selectively retain global structure and filter out noisy or misleading local signals.
> ## Q3: ##
> We thank the reviewer for pointing out the importance of the spectral truncation parameter $r$ in FEDLAP+. This parameter controls how many eigenvectors of the graph Laplacian are retained during the spectral decomposition step.
>
> More precisely, the global Laplacian matrix $L_G$ is approximated by the low-rank decomposition $L_G \approx U_{[:, r]} \Lambda_{[r, r]} U_{[:, r]}^\top$, where $U_{[:, r]} \in \mathbb{R}^{n \times r}$ contains the first $r$ eigenvectors (corresponding to the smallest eigenvalues), and $\Lambda_{[r, r]}$ is the diagonal matrix of the corresponding eigenvalues. The matrix $W \in \mathbb{R}^{r \times d_s}$ is then learned in the spectral domain. Truncating to the top-$r$ smooth eigenvectors serves three purposes:
> - It reduces communication (only the top $r$ eigenvectors are shared),
> - It improves privacy (smaller $r$ reduces potential leakage),
> - It acts as a regularizer, filtering out high-frequency components (which often capture noise) that may harm generalization.
>
> To choose $r$ in practice:
> - A rule of thumb is to set $r$ close to the intrinsic rank of the graph, which often corresponds to the number of clusters or communities. For many real-world datasets, this number is much smaller than the total number of nodes $n$.
> - In our experiments (see Fig.5 and Fig.6 of the paper), we observed that moderate values of $r$ (e.g., 50–200) work well across all datasets. For example, in PubMed (with over 19K nodes), $r = 80$ already gives strong performance, and further increasing $r$ yields diminishing returns.
> - We also show in Fig.5 (left panel) and Fig.6 that the performance of FEDLAP+ is robust across a wide range of $r$ values,
>
> We will include this detailed explanation of the truncation parameter $r$ in the revised manuscript.
>
> ## Q4: ##
> Thank you for the insightful question. While FEDLAP+ is designed for static graphs, its use of decentralized Arnoldi iteration makes it amenable to handling dynamic graphs efficiently. In particular, when the graph structure evolves incrementally (e.g., through the addition or removal of a few nodes or edges), one can apply Krylov subspace recycling [1] to update the spectral components without recomputing them from scratch. This makes it possible to adapt to dynamic settings with minimal communication overhead.
>
> Our current framework, like all other existing SFL methods (e.g., FEDSTRUCT, FEDGCN),
> focuses on fixed graphs. However, we are currently investigating the extension of FEDLAP+ to dynamic graphs as part of our ongoing research. We believe that developing privacy-preserving SFL methods for evolving graphs is a promising and largely unexplored direction for future work.
> [1] Bolten et al., "Krylov subspace recycling for evolving structures," Computer Methods in Applied Mechanics and Engineering, 2022.
> ## Q5:  ##
> Indeed, FEDSTRUCT achieves stronger performance than FEDLAP on Chameleon, and we appreciate the reviewer’s observation. Note that Chameleon is a heterophilic graph where neighbors often belong to different classes and have dissimilar features. In such settings, Laplacian-based smoothing, whether applied in the spatial domain (as in FEDLAP) or the spectral domain (as in FEDLAP+), can propagate misleading signals across class boundaries. Although Laplacian smoothing has known limitations on heterophilic graphs, FEDLAP+ overcomes this issue by truncating high-frequency spectral components that often carry noisy signals. As a result, it achieves competitive performance, often close to state-of-the-art models, despite its privacy-preserving design.
>
> We reiterate that we do not position FEDLAP as a standalone solution for heterophilic graphs, but as a step toward FEDLAP+, which is the primary contribution of our work. FEDLAP+ achieves performance very close to FEDSTRUCT on Chameleon.
>
> Hybrid approaches that combine spectral and feature-level smoothing could potentially improve performance further. However, they typically require additional cross-client information sharing, which increases the risk of privacy leakage. In contrast, FEDLAP and FEDLAP+ are explicitly designed to limit the information exchanged to minimize such leakage. It is also worth noting that FEDSTRUCT shares significantly more information across clients and does not offer formal privacy guarantees. Its slightly stronger performance on Chameleon with respect to FEDLAP+ likely results from this additional information exchange, which comes at the cost of increased privacy exposure.

---

> > ### Author Response · Authors · 2025-08-06
> >
> > Dear Reviewer,
> >
> > We believe we have thoroughly and satisfactorily addressed all of your comments in our response. We would greatly appreciate it if you could take a moment to review our replies and let us know if any concerns remain. We're happy to clarify any points further. If our responses resolve your concerns, we would be grateful if you would consider updating your score accordingly.
> >
> > Thank you,
> > The authors

---

### Note · Authors · 2025-08-12

Our work presents FedLAP+, a spectral-domain framework for subgraph FL (SFL) that achieves competitive or superior accuracy to SOTA while providing rigorous, quantifiable privacy. FedLAP+ separates SFL into 1) an offline spectral compression phase, where structural knowledge is privately extracted, and 2) a standard FL online phase.

The original reviews were overall positive: HaMS, 3cQg, Ea2b (score 4) all appreciated our contribution. 7gcZ (score 2) was the only critical reviewer.

Discussion outcome: We believe we addressed all reviewers’ points thoroughly and convincingly. 3cQg and Ea2b were satisfied with our rebuttal, HaMS did not respond, and 7gcZ was happy with parts of our rebuttal, raised the score, but didn’t detail which concerns remained.

7gcZ requested to “expand a bit more on the privacy” and justify that our analysis provides “strong privacy”. We clarified that privacy concerns apply only to the offline phase, since the online phase is identical to standard FL and can adopt protections like DP or SecAgg. In the offline phase, only structural information (no node features) is exchanged. Thus, our privacy analysis focuses on leakage of structural information, ie, revealing edges. Our approach is grounded in membership inference-based auditing. We believe we supported why focusing on edge membership yields strong privacy and will add clarifications and the link to membership inference attacks in the paper. 7gcZ raised the score, expressed some concerns remained, but didn’t specify, and we were’nt given the opportunity to clarify those.

HaMS and 3cQg demanded more justification of the privacy analysis assumptions, which we provided: We deliberately model an unrealistically strong attacker to ensure robustness. The attack is formalized as an optimal hypothesis test, with closed-form KL bounds derived and validated analytically and empirically. Results show the attacker’s success is worse than trivial guessing, even in this worst-case scenario (3cQg was satisfied).

Other remarks: Ea2b asked about finite-n effects; we showed the error decreases rapidly with n (Ea2b was satisfied). Ea2b queried the Erdos-Renyi assumption; it models attacker uncertainty and yields general bounds, validated on fixed real graphs. FedLAP+ is task-agnostic (node/edge level). Efficiency (HaMS, 7gcZ): one-time offline Arnoldi (linear), online overhead is negligible. Performance (HaMS, 3cQg): matches or beats SOTA on homo/hetero graphs with strong privacy and low cost.

---

### Decision · Program_Chairs · 2025-09-17

**Decision:**

Accept (poster)

**Comment:**

The reviewers unanimously recognized the novel methodological contribution of this paper, which introduces a spectral-domain framework for private, structure-aware subgraph federated learning.

While initial reviews raised concerns about performance in heterophilic settings (3cQg, HaMS) and the assumptions for the privacy proofs (Ea2B, 3cQg), the authors’ rebuttal successfully addressed most of these issues. The primary remaining point of contention was the rigor of the theoretical analysis, a point that Reviewers Ea2b and 7gcZ felt was not fully resolved. They noted that the proof for the online component was missing and that the theorem on privacy guarantees lacked clarity and formal precision.

Despite this weakness in the theoretical presentation, the paper’s core contribution is significant and it is free of critical flaws. Therefore, I recommend acceptance. I strongly encourage the authors to address the feedback on the theoretical proofs in the final version, either by strengthening them or by moderating their claims of rigor.